# Be a Goldfish: Forgetting Bad Conditioning in Sparse Linear Regression via Variational Autoencoders

**Kuheli Pratihar** [1]   **Debdeep Mukhopadhyay** [1]

## Abstract

Variational Autoencoders (VAEs), a class of latent-variable generative models, have seen extensive use in high-fidelity synthesis tasks, yet their loss landscape remains poorly understood. Prior theoretical works on VAE loss analysis have focused on their latent-space representational capabilities, both in the optimal and limiting cases. Although these insights have guided better VAE designs, they also often restrict VAEs to problem settings where classical algorithms, such as Principal Component Analysis (PCA), can trivially guarantee globally optimal solutions. In this work, we push the boundaries of our understanding of VAEs beyond these traditional regimes to tackle NP-hard sparse inverse problems, for which no classical algorithms exist. Specifically, we examine the nontrivial Sparse Linear Regression (SLR) problem of recovering optimal sparse inputs in the presence of an ill-conditioned design matrix having correlated features. We provably show that, under a linear encoder-decoder architecture incorporating the product of the SLR design matrix with a trainable, sparsity-promoting diagonal matrix, any minimum of VAE loss is guaranteed to be an optimal solution. This property is especially useful for identifying (a) a preconditioning factor that reduces the eigenvalue spread, and (b) the corresponding optimal sparse representation. Lastly, our empirical analysis with different types of design matrices validates these findings and even demonstrates a higher recovery rate at low sparsity where traditional algorithms fail. Overall, this work highlights the flexible nature of the VAE loss, which can be adapted to efficiently solve computationally hard problems under specific constraints.

[1]Department of Computer Science and Engineering, Indian Institute of Technology Kharagpur, Kharagpur, India. Correspondence to: Kuheli Pratihar <its.kuheli96@gmail.com>.

*Proceedings of the $42^{nd}$ International Conference on Machine Learning*, Vancouver, Canada. PMLR 267, 2025. Copyright 2025 by the author(s).

## 1. Introduction

Variational Autoencoders (VAEs) (Kingma & Welling, 2014) excel at modeling complex, unknown distributions of observed data. Their ability to capture complex latent structures makes them particularly effective for high-fidelity image synthesis (Gulrajani et al., 2017), text generation in natural language processing (NLP) (Serban et al., 2017), and forecasting new data points in time-series analysis (Löwe et al., 2022). Despite these successes, the theoretical underpinnings of VAEs remain only partially understood, leaving open questions about their full potential.

Recent theoretical developments have primarily examined the latent space representational capabilities of VAEs (Zheng et al., 2022; Dai et al., 2018; 2021), which support high-quality reconstructions. However, when the observed data lies in a low-dimensional space, an affine VAE's latent representation effectively reduces to probabilistic PCA (Wipf, 2023). Under these conditions, there is little difference between a VAE and a deterministic autoencoder (AE), as both can learn the principal subspace of the data equally well (Dai et al., 2018; Lucas et al., 2019). Consequently, the primary value of these analyses lies in elucidating properties of the underlying energy functions, which guide the design of VAEs capable of accurately learning data representations. Nonetheless, they constrain our perspective on the broader capabilities that VAEs may offer.

Confirming these observations, recent work (Wipf, 2023) leverages VAEs to solve an NP-hard sparse inverse problem—an area in which generative models remain largely unexplored. In particular, (Wipf, 2023) finds the optimal solution for Simultaneous Sparse Regression (SSR), a task that conventional algorithms typically fail to address reliably. This success stems from a remarkable property: under specific encoder-decoder architectures, VAEs exhibit no local minima. Consequently, all global minima correspond to the optimal solutions, providing a maximally sparse representation for SSR. These findings motivate the objective of *broadening our understanding of VAEs as tools for solving NP-hard problems under non-trivial conditions where existing algorithms fail to yield reliable solutions.*

In this work, we focus on Sparse Linear Regression (SLR) (Donoho & Stark, 1989), a widely studied problem in

high-dimensional statistics. The goal is to identify the optimal sparse solution for a system of linear equations based on observed data. The key challenge in SLR lies in identifying the sparse solution with the minimum error from a combinatorial set of potential solutions. From an optimization perspective, SLR features a non-convex $\ell_0$-norm constraint coupled with a mean squared error objective, leading to a large number of suboptimal local minima and rendering the problem NP-hard. In this context, the ability of VAEs to eliminate spurious local minima becomes pivotal for recovering the optimally sparse solution.

Specifically, we consider two prominent non-trivial scenarios in which the state-of-the-art SLR algorithm, LASSO, provably fails (Kelner et al., 2022b). The challenge of identifying an optimal sparse solution among a combinatorial set of suboptimal local minima is exacerbated when (a) the design matrix is ill-conditioned with highly correlated columns, or (b) the ground-truth SLR coefficients are not sparse. Such conditions commonly arise in real-world settings, including signal-processing and compressed sensing (Rudelson & Vershynin, 2006; Hassanieh et al., 2012), as well as feature selection tasks in privacy-preserving machine learning (PPML) (Akavia et al., 2024; Li et al., 2021), thereby making the search for an optimal solution substantially more challenging.

For an ill-conditioned design matrix, preconditioning (Kelner et al., 2022b; Wauthier et al., 2013) is often employed to improve its condition number. However, finding an appropriate preconditioner is a non-trivial task, and efficient algorithms do not exist. In this context, we ask:

*How can we adapt VAEs to intrinsically precondition ill-conditioned design matrices?*

While existing results show that VAEs with specific encoder-decoder architectures can provably achieve optimal sparse solutions for SSR (Wipf, 2023), they do not address the condition number of the design matrix. In this work, we propose a VAE architecture that intrinsically reduces the eigenvalue spread for any arbitrary full-rank fat design matrix, thereby preconditioning it. Consequently, design matrices whose condition number improves via this reduced eigenvalue spread can achieve the optimal SLR solution using the proposed VAE architecture. Next, in the case of a large number of non-sparse coefficients, the most common strategy is to increase the number of observations in the SLR problem (Wainwright, 2006). However, classical algorithms typically fail when the level of sparsity is too low, leading to the question:

*Can VAEs intrinsically extend the sparsity threshold for SLR with a fixed number of observations?*

Through empirical evaluations of various design matrices,

we observe that the answer is affirmative. We conjecture that the VAE's inherent sparsity-inducing mechanisms contribute to this improved tolerance for low-sparsity scenarios.

In this work, we leverage the ability of VAEs to eliminate spurious local minima for the non-trivial settings of SLR. To summarize, our main contributions are:

- **Optimal Sparse Solution for SLR using VAEs:** We show, for the first time, that VAEs can provably identify the optimal sparse solution for well-conditioned design matrices. This result relies on a VAE architecture with a linear encoder and decoder, both equipped with a sparsity-promoting diagonal matrix that eliminates local minima, thereby ensuring convergence to a global minimum.

- **Preconditioning for Ill-Conditioned SLR:** We demonstrate that VAEs inherently reduce the eigenvalue spread of any full-rank fat design matrix, effectively preconditioning it. In cases when the conditioning of the design matrix improves through this mechanism, VAEs achieve the optimal SLR solution.

- **Greater Tolerance to Low-Sparsity:** With the proposed encoder-decoder architecture, VAEs maintain a high recovery rate of sparse indices even at lower sparsity levels. Notably, we empirically demonstrate that for various types of design matrices, VAEs achieve greater tolerance to low sparsity at a fixed number of observations. In contrast, conventional algorithms such as LASSO and augmented basis pursuit fail to perform well at lower sparsity levels.

The remainder of this paper is organized as follows. Section 2 provides a detailed overview of the theoretical underpinnings of VAEs. In Section 3, we discuss the fundamentals of SLR and its challenges. Section 4 begins by proving the existence of an optimal SLR solution using a VAE in the well-conditioned case, followed by our proposed VAE architecture for preconditioning SLR. Section 5 details our experimental results for different types of design matrices, and finally, Section 6 concludes the paper.

## 2. Background and Related Works

### 2.1. Variational Autoencoder

Formally, VAEs achieve this by training a $\theta$-parameterized marginal distribution $p_\theta(\mathbf{x})$, defined as:

$$p_\theta(\mathbf{x}) \;=\; \int p_\theta(\mathbf{x}, \mathbf{z}) \, d\mathbf{z} \qquad (1)$$

where $\mathbf{x} \in \mathbb{R}^d$ is the $d$-dimensional observed data, $\mathbf{z} \in \mathbb{R}^n$ is $n$-dimensional unobserved latent variable. In scenarios

where $\mathbf{x}$ has low dimension, one often enforces $n < d$ to limit the number of latent dimensions. However, directly computing (1) is typically not feasible because the joint distribution $p_\theta(\mathbf{x}, \mathbf{z})$, which is also needed to find the true posterior $p_\theta(\mathbf{z} \mid \mathbf{x})$, is intractable in most practical settings. Consequently, VAEs employ *Variational Inference* (VI), which recasts the approximation of the ground-truth distribution as an optimization problem by replacing the intractable posterior $p_\theta(\mathbf{z} \mid \mathbf{x})$ with a simpler distribution $q_\phi(\mathbf{z} \mid \mathbf{x})$ (often Gaussian) and minimizing their Kullback–Leibler (KL) divergence between them.

To perform this optimization, VAEs adopt an autoencoder-style architecture in which the encoder, parameterized by $\phi$, approximates the posterior $q_\phi(\mathbf{z} \mid \mathbf{x})$, and the decoder, parameterized by $\theta$, models the likelihood $p_\theta(\mathbf{x} \mid \mathbf{z})$. In practice, both are often chosen to be Gaussian:

$$p_\theta(\mathbf{x} \mid \mathbf{z}) = \mathcal{N}\big(\mathbf{x} \mid \mu_x(\mathbf{z};\theta), \gamma I\big),$$
$$q_\phi(\mathbf{z} \mid \mathbf{x}) = \mathcal{N}\big(\mathbf{z} \mid \mu_z(\mathbf{x};\phi), \Sigma_z(\mathbf{x};\phi)\big), \qquad (2)$$

together with the prior $p(\mathbf{z}) = \mathcal{N}\big(\mathbf{z} \mid 0, I\big)$. Here, $\gamma > 0$ is a (trainable or fixed) scalar variance, and the functions $\mu_x(\mathbf{z};\theta)$, $\mu_z(\mathbf{x};\phi)$, and $\Sigma_z(\mathbf{x};\phi)$ are instantiated by neural networks. With the above modeling, the KL-divergence between intractable posterior $p_\theta(\mathbf{z}|\mathbf{x})$ with a simpler distribution $q_\phi(\mathbf{z}|\mathbf{x})$ can be simplified to obtain VAE loss $\mathcal{L}(\phi, \theta)$ given by:

$$-\mathbb{E}_{q_\phi(\mathbf{z}|\mathbf{x})}\big[\log p_\theta(\mathbf{x} \mid \mathbf{z})\big] + \mathrm{KL}\big[q_\phi(\mathbf{z} \mid \mathbf{x}) \,\|\, p(\mathbf{z})\big]. \quad (3)$$

The first term in this expression is the reconstruction loss, while the second is the regularization term expressed as the KL divergence between the learned latent distribution and the prior. The loss in (3) is then optimized over $(\phi, \theta)$ on the training data using stochastic gradient descent (SGD) to find the optimal latent representation or to generate synthetic samples from the input.

### 2.2. VAE Global Minima Analysis

Prior theoretical analyses of VAEs have shown that global minima of the VAE objective can indeed recover the underlying data manifold (Zheng et al., 2022; Koehler et al., 2022; Dai et al., 2017; Lucas et al., 2019), demonstrating their excellence in approximating the ground-truth distribution. However, reaching these global minima is often delegated to an optimization algorithm such as SGD, which can be hindered by spurious local minima in the VAE loss landscape. Interestingly, it has been shown that certain architectural modifications to VAEs can eliminate these bad local minima, thereby facilitating convergence to the global optimum. First conjectured in (Dai & Wipf, 2019) and later proven in (Wipf, 2023), marginalizing over the posterior distribution effectively smooths away spurious minima that exploit an excessive number of latent dimensions to reduce

reconstruction error. Moreover, the global minima of such a marginalization-based VAE loss correspond to the optimal sparse representation for the NP-hard Simultaneous Sparse Regression (SSR) problem, where no polynomial-time algorithm is known. This smoothing mechanism thus helps eliminate the exponential number of suboptimal sparse solutions in SSR that arise for a given sparsity level. Several other studies have also examined the optimization trajectory of VAEs (Dai et al., 2018; Shekhovtsov et al., 2022; Zietlow et al., 2021; Damm et al., 2023). These insights pave the way for designing VAE architectures that remove bad local minima in other sparse inverse problems, such as sparse linear regression (SLR), the primary focus of our work.

In the next section, we elaborate on the challenges of solving SLR that motivate the construction of our proposed VAE technique.

## 3. Challenges in Solving Sparse Linear Regression

In *sparse linear regression* (SLR), we are given a design matrix $\boldsymbol{\Phi} \in \mathbb{R}^{d \times n}$ and observations $\mathbf{x} \in \mathbb{R}^d$ satisfying

$$\mathbf{x} = \boldsymbol{\Phi}\mathbf{z}^* + \boldsymbol{\eta}, \qquad (4)$$

where $\mathbf{z}^* \in \mathbb{R}^n$ is $\kappa$-*sparse* coefficient vector, meaning it has at most $\kappa$ non-zero entries, and $\boldsymbol{\eta} \in \mathbb{R}^d$ is a small noise term. This is a hard problem compared to standard linear regression, where there are no constraints on the sparsity of $\mathbf{z}^*$ and can be optimally solved using the ordinary least squares (OLS) algorithm. The SLR hardness arises from the difficulty in searching through an exponential solution space where both the location and the value of the sparse coefficients are unknown. Our goal is to find a $\kappa$-sparse $\hat{\mathbf{z}}$ that minimizes the $\ell_2$ error with the sparsity constraint:

$$\hat{\mathbf{z}} = \arg\min_{\mathbf{z}:\, \|\mathbf{z}\|_0 = \kappa} \|\mathbf{x} - \boldsymbol{\Phi}\mathbf{z}\|_2^2. \qquad (5)$$

Because the $\ell_0$ constraint makes (5) nonconvex, a frequent strategy is to add a sparsity-inducing penalty term $g(z_i)$ for each coefficient:

$$\hat{\mathbf{z}} = \arg\min_{\mathbf{z}} \|\mathbf{x} - \boldsymbol{\Phi}\mathbf{z}\|_2^2 + \lambda \sum_{i=1}^{n} g(z_i), \qquad (6)$$

where $\lambda > 0$ is a trade-off parameter. A widely used choice for $g(z_i)$ is the $\ell_1$ norm, which gives the LASSO (Tibshirani, 1996), i.e. $\sum_i g(z_i) = \|\mathbf{z}\|_1$.

However, the performance of LASSO depends on the conditioning of the design matrix $\boldsymbol{\Phi}$. In particular, a random $\boldsymbol{\Phi}$ must often satisfy a small *Restricted Isometry Constant* (RIC) (Candes & Tao, 2005), which ensures that $\boldsymbol{\Phi}$ behaves nearly like an orthonormal system on every $\kappa$-sparse subset of coefficients (see below for definition). Concretely, no set

of $\kappa$ columns in $\boldsymbol{\Phi}$ is nearly linearly dependent, so no sparse vector $\mathbf{z}$ is "collapsed" or "inflated" by $\boldsymbol{\Phi}$.

**Definition 3.1** (Restricted Isometry Constant). Let $\boldsymbol{\Phi} \in \mathbb{R}^{d \times n}$ be a real matrix. For an integer $\kappa \leq n$, the $\kappa$-*Restricted Isometry Constant* $\delta$ of $\boldsymbol{\Phi}$ is the smallest non-negative number such that,

$$(1 - \delta)\|\mathbf{z}\|_2^2 \;\leq\; \|\boldsymbol{\Phi}\mathbf{z}\|_2^2 \;\leq\; (1 + \delta)\|\mathbf{z}\|_2^2 \quad (7)$$

for *every* vector $\mathbf{z} \in \mathbb{R}^n$ that has at most $\kappa$ nonzero entries. We abbreviate this condition as *Restricted Isometry Property* (RIP) for future use.

One can equivalently interpret this definition in terms of singular values of all $\kappa$-column submatrices of $\boldsymbol{\Phi}$: a small $\delta$ forces those submatrices to be well-conditioned.

### 3.1. Requirements for Well-conditioning

A helpful viewpoint is to examine the (scaled) "Gram matrix" $\boldsymbol{\Sigma} = \frac{1}{d}\boldsymbol{\Phi}^\top\boldsymbol{\Phi} \in \mathbb{R}^{n \times n}$. This Gram matrix perspective also arises when the rows of $\boldsymbol{\Phi}$ are sampled i.i.d. from a distribution with covariance $\boldsymbol{\Sigma}$. In that setting, $\boldsymbol{\Sigma} = \frac{1}{d}\boldsymbol{\Phi}^\top\boldsymbol{\Phi}$ serves as the empirical (sample) covariance. If $\boldsymbol{\Phi}$ satisfies the RIP with $\delta$ sufficiently small (e.g. $\delta < 1$), then for every $\kappa$-sparse vector $\mathbf{z} \in \mathbb{R}^n$,

$$\|\boldsymbol{\Phi}\mathbf{z}\|_2^2 \;=\; d\mathbf{z}^\top\left(\frac{1}{d}\boldsymbol{\Phi}^\top\boldsymbol{\Phi}\right)\mathbf{z} \;=\; d\left[\mathbf{z}^\top\boldsymbol{\Sigma}\mathbf{z}\right]; \quad (8)$$

$$(1 - \delta)\|\mathbf{z}\|_2^2 \;\leq\; d\left[\mathbf{z}^\top\boldsymbol{\Sigma}\mathbf{z}\right] \;\leq\; (1 + \delta)\|\mathbf{z}\|_2^2. \quad (9)$$

Hence, all eigenvalues $\sigma$ of $\boldsymbol{\Sigma}$ lie in the interval $[\,1 - \delta,\; 1 + \delta\,]$, which implies,

$$\operatorname{cond}(\boldsymbol{\Sigma}) \;=\; \frac{\sigma_{\max}(\boldsymbol{\Sigma})}{\sigma_{\min}(\boldsymbol{\Sigma})} \;\leq\; \frac{1 + \delta}{1 - \delta}, \quad (10)$$

where $\operatorname{cond}(.)$ is the condition number. Thus, a small RIC $\delta$ guarantees not only that $\boldsymbol{\Phi}$ acts almost as an isometry on all $\kappa$-sparse vectors, but also that the Gram matrix $\boldsymbol{\Sigma}$ is *well-conditioned*.

### 3.2. Preconditioning for Ill-Conditioned Design Matrices

When $\boldsymbol{\Phi}$ does not satisfy the RIP, or equivalently when $\operatorname{cond}(\boldsymbol{\Sigma})$ is large (indicating ill-conditioning typically caused by highly correlated columns), finding an optimal $\kappa$-sparse solution is generally computationally intractable (Kelner et al., 2022a). Nonetheless, methods such as LASSO and other sparse estimators can often perform well in practice if $\boldsymbol{\Sigma}$ is sufficiently well-conditioned (Gupte et al., 2024; Kelner et al., 2022a).

A standard approach for improving conditioning is *preconditioning*: we multiply both $\boldsymbol{\Phi}$ and $\mathbf{x}$ by a matrix $\mathbf{P} \in \mathbb{R}^{n \times n}$. The resulting preconditioned SLR is given by:

$$\mathbf{P}\mathbf{x} \;=\; \mathbf{P}\boldsymbol{\Phi}\mathbf{z}^* + \mathbf{P}\mathbf{w} \implies \tilde{\mathbf{x}} \;=\; \tilde{\boldsymbol{\Phi}}\mathbf{z}^* + \tilde{\mathbf{w}}, \quad (11)$$

where $\tilde{\boldsymbol{\Phi}} \equiv \mathbf{P}\boldsymbol{\Phi}$, $\tilde{\mathbf{x}} \equiv \mathbf{P}\mathbf{x}$, and $\mathbf{z}^*$ remains the same $\kappa$-sparse solution as in (4). The key requirement is that $\tilde{\boldsymbol{\Phi}}$ be better conditioned (exhibit a smaller $\delta$ and a bounded condition number), thus enabling LASSO or related algorithms to recover the optimal sparse representation more reliably.

However, constructing a suitable preconditioner can be challenging and often relies on specific properties of $\boldsymbol{\Phi}$, such as low treewidth in a Markov structure (Kelner et al., 2022a). Moreover, (Gupte et al., 2024) shows that there exist ill-conditioned $\boldsymbol{\Phi}$ for which no polynomial-time algorithm can produce a preconditioner sufficient to achieve the optimal solution, implying an average-case hardness for SLR.

### 3.3. Sample Complexity in Solving SLR

Although LASSO is the de facto algorithm for sparse linear regression (SLR) when the design matrix is well-conditioned, it requires on the order of $d = \Omega(\kappa \log n)$ observations to reliably recover the optimal solution (Kelner et al., 2024; 2022b). Consequently, if the number of non-zero elements $\kappa$ is large and the number of samples is insufficient, LASSO fails. In fact, (Zhang et al., 2011) shows that solving SLR with lower sparsity levels (*i.e.*, fewer zero entries) can be significantly more challenging than the case of higher sparsity. Confirming this limitation of LASSO, prior works on preconditioning SLR generally target higher sparsity levels (Kelner et al., 2022b; Wainwright, 2006; Jia & Rohe, 2015). However, in this work, we also consider the regime of lower sparsity.

Next, we describe how VAEs can help overcome these challenges in solving SLR despite ill-conditioning and an unfavorable sparsity regime.

## 4. Optimal SLR Solution using VAE

Leveraging the bad local minima smoothening property of VAEs (Wipf, 2023), we first demonstrate how VAEs, constructed with a specific encoder-decoder architecture, can achieve an optimal sparse representation for SLR (6). We do so in the noise-free scenario when the design matrix $\boldsymbol{\Phi}$ is well-conditioned. Thereafter, for an ill-conditioned $\boldsymbol{\Phi}$ we propose a VAE architecture with intrinsic preconditioning that reduces the spread in the eigenvalues, leading to a well-conditioned $\tilde{\boldsymbol{\Phi}}$ under certain conditions.

We define *optimal sparse representations* as such that $\mathbf{x} = \boldsymbol{\Phi}\mathbf{z}$ with $\kappa$-sparse $\mathbf{z}$ achieves zero reconstruction error:

$$\|\mathbf{x} - \boldsymbol{\mu}_x[\boldsymbol{\mu}_z(\mathbf{x}; \phi); \theta]\|_2^2 \;=\; 0, \quad (12)$$

where $\boldsymbol{\mu}_z$ is the encoder output and $\boldsymbol{\mu}_x$ is the decoder output for VAE (2). This condition, described in (Dai et al., 2021), is admittedly restrictive but is nonetheless employed as the search objective in many sparse-inverse problems (Candes & Tao, 2005; Candès et al., 2006). Thereafter, we use the

following lemma from (Dai & Wipf, 2019) to ascertain the sparse representational properties of VAE:

**Lemma 4.1.** *Assume a Gaussian VAE model of continuous data defined by* (2), *where* $\boldsymbol{\mu}_x = \mathbf{W}_x \mathbf{z} + \mathbf{b}_x$ *for some weight matrix* $\mathbf{W}_x$ *and bias vector* $\mathbf{b}_x$; *similarly,* $\boldsymbol{\mu}_z = \mathbf{W}_z \mathbf{x} + \mathbf{b}_z$ *for some weight matrix* $\mathbf{W}_z$ *and bias vector* $\mathbf{b}_z$ *and* $\boldsymbol{\Sigma}_z = diag[\mathbf{s}]^2$, *where* $\mathbf{s}$ *is an arbitrary parameter vector independent of* $\mathbf{x}$. *Then for any fixed value of* $\gamma$, *all local minima of the resulting VAE objective with respect to the parameters* $\{\hat{\mathbf{W}}_x, \hat{\mathbf{b}}_x, \hat{\mathbf{W}}_z, \hat{\mathbf{b}}_z, \hat{\mathbf{s}}\}$ *are also global minima. Moreover, these global minima produce* optimally sparse *representations* (12), *when* $\gamma \to 0$.

This lemma highlights how linear encoder-decoder architectures can attain sparse latent representations when the data lies on a low-dimensional manifold. Intuitively, an optimal sparse representation occurs because each local/global minimum selects a principal subspace of the data while using the fewest possible nonzero columns of $\mathbf{W}_x$. Furthermore, at the indices of those zero-valued columns, elements of $\mathbf{s}$ tend to zero as $\gamma \to 0$. In contrast, the corresponding elements of $\boldsymbol{\mu}_z$ convey the information about $\mathbf{x}$ (i.e., the active, non-random dimensions) needed for exact data reconstruction. Next, we propose a VAE architecture, following Lemma 4.1, for the SLR objective (6).

The SLR objective is closely tied to learning a data-specific latent prior $p(\mathbf{z})$ in variational Bayes (VB) methods (Wipf et al., 2011). However, in contrast to VB methods that model the prior distribution on the latent space, we propose encoding the "sparsity" information via a trainable diagonal matrix in the VAE decoder, assuming a standard Gaussian as the latent prior. We later demonstrate that such an encoding is beneficial for learning the optimal sparse representation (12) as opposed to finding the optimal latent prior for VB methods.

The resulting decoder's Gaussian distribution $p_\theta(\mathbf{x}|\mathbf{z}) = \mathcal{N}(\mathbf{x}; \boldsymbol{\Phi} \operatorname{diag}[\mathbf{w}_x] \mathbf{z}, \gamma \mathbf{I})$ is parameterized by: $\boldsymbol{\mu}_x = \boldsymbol{\Phi} \operatorname{diag}[\mathbf{w}_x] \mathbf{z}$, and $\boldsymbol{\Sigma}_x = \gamma \mathbf{I}$, where $\operatorname{diag}[\mathbf{w}_x]$ is a sparsity-promoting diagonal matrix which selects the non-zero sparse features from $\boldsymbol{\Phi}$ implicitly during the training process. Next, for the encoder, we use a linear mean vector $\boldsymbol{\mu}_z = \mathbf{W}_z \mathbf{x}$ where $\mathbf{W}_z \in \mathbb{R}^{n \times d}$ and a full-rank covariance $\boldsymbol{\Sigma}_z = \mathbf{S}\mathbf{S}^\top$, with $\mathbf{S} \in \mathbb{R}^{n \times n}$ arbitrary and independent of $\mathbf{x}$. Note that, we assume $\boldsymbol{\Phi} \in \mathbb{R}^{d \times n}$ is a *fat*, full-rank matrix, i.e. $d < n$ and $\operatorname{rank}(\boldsymbol{\Phi}) = \min(d, n) = d$. This full-rank constraint ensures that $\boldsymbol{\Phi}$ has $d$ nonzero eigenvalues.

With these choices, the VAE energy from (2), applied to the training data $\mathbf{x}$, reduces to:

$$\mathcal{L}(\theta, \phi) = \mathbb{E}_{q_\phi(\mathbf{z}|\mathbf{x})}\left[\frac{1}{\gamma}\left\|\mathbf{x} - \boldsymbol{\Phi} \operatorname{diag}[\mathbf{w}_x] \mathbf{z}\right\|^2\right] + d\log\gamma$$
$$+ \operatorname{tr}[\mathbf{S}\mathbf{S}^\top] - \log|\mathbf{S}\mathbf{S}^\top| + \|\mathbf{W}_z \mathbf{x}\|_2^2, \quad (13)$$

where $\theta = \{\mathbf{w}_x, \gamma\}$ and $\phi = \{\mathbf{W}_z, \mathbf{S}\}$.

We are now positioned to show that, when the design matrix $\boldsymbol{\Phi}$ is well-conditioned (i.e. there are no closely correlated columns), the loss function in (13) does not admit any "bad" local minima.

**Theorem 4.2.** *Any local minimum of* (13), $\{\hat{\mathbf{w}}_x, \hat{\mathbf{W}}_z, \hat{\mathbf{S}}\}$ *with a fixed* $\gamma$, *is also a global minimum. When* $\gamma \to 0$, *and* $\boldsymbol{\Phi}$ *is well-conditioned, satisfying the RIP condition for small* $\delta$, *the global minima achieve the optimal sparse solution for SLR in* (4). *The resulting sparse coefficients are given by:*

$$\hat{\mathbf{z}} = \operatorname{diag}[\hat{\mathbf{w}}_x]^2 \boldsymbol{\Phi}^\top \left(\boldsymbol{\Phi} \operatorname{diag}[\hat{\mathbf{w}}_x]^2 \boldsymbol{\Phi}^\top\right)^{-1} \mathbf{x}. \quad (14)$$

For brevity, the proof has been deferred to the Appendix A. We first show that (13) has no bad local minima, under the assumption that $\boldsymbol{\Phi}$ is well-conditioned or satisfies the $(\kappa, \delta)$-RIP with $\delta < 1$. With a perfect optimizer, a full rank $\boldsymbol{\Phi}$ ensures the presence of a unique inverse $\Sigma^{-1}(\mathbf{w})$ for each $\mathbf{w}$, leading to no bad local minima. However, practical SGD might identify distinct $\mathbf{w}$'s with close inverse values. We use the RIP bound $\delta$ to ensure that the difference in inverse terms is large enough to be detectable by SGD. For a $\kappa$-sparse z, the RIP bound $\delta$ can be expressed as a weighted sum of activated column norms of $\boldsymbol{\Phi}$ and cross correlations between them. However, the presence of aligned columns leads to large correlations, increasing the $\delta$ value. This reduces the difference in inverse terms for distinct $\mathbf{w}$'s that differ along the indices of the aligned columns, leading to the requirement of a small $\delta$ to find the true optimum. Thereafter, using Lemma 4.1, we show that for such a well-conditioned $\boldsymbol{\Phi}$, this global minimum of (13) coincides with the *optimal sparse solution* in the limit $\gamma \to 0$.

**Interpretation of Theorem 4.2:** The key takeaway for Theorem 4.2 is that one can construct a VAE architecture for SLR with a trainable diagonal matrix to enforce sparsity *rather than* relying on an explicit $\ell_1$ penalty (as in LASSO). Moreover, the VAE-based approach, grounded in variational inference, can learn both the mean ($\mathbf{W}_z$) and covariance ($\mathbf{S}\mathbf{S}^\top$) of the sparse coefficients, thus embedding information about both the number of nonzero coefficients and their distribution. This is distinct from classical $\ell_1$ regularization algorithms such as LASSO, which focus on constraining the number of nonzero coefficients but do not explicitly model their distribution.

Nonetheless, the requirement that $\boldsymbol{\Phi}$ be well-conditioned motivates further consideration of how VAEs can be adapted for ill-conditioned dictionaries, which we address next.

### 4.1. Preconditioning using VAEs

In order to achieve the loss-smoothing property for (13), the columns of $\boldsymbol{\Phi}$ must be non-collinear. When $\boldsymbol{\Phi}$ is ill-conditioned, its columns are nearly linearly dependent, lead-

ing to multiple local minima with distinct solutions in (13). Ill-conditioning can be reflected in various properties, such as failing the $(\kappa, \delta)$- RIP for smaller $\delta$ or having a large condition number. Here we focus on reducing the condition number of $\boldsymbol{\Phi}$ using VAEs, as it is directly tied to the spread of its eigenvalues. The well-known ridge regularizer (Hoerl & Kennard, 1970) provides a way to shrink this spread for a given $\gamma$ via the *preconditioning factor*:

$$\mathbf{P} = (\boldsymbol{\Phi}\,\boldsymbol{\Phi}^{\top} + \gamma\,\mathbf{I})^{-1} \in \mathbb{R}^{d \times d}. \quad (15)$$

While conventionally employed to boost the eigenvalues of low-rank matrices, we use (15) to reduce the difference between the largest and smallest eigenvalues of the (fat) full-rank matrix $\boldsymbol{\Phi}$.

**Lemma 4.3.** *Let* $\boldsymbol{\Phi} \in \mathbb{R}^{d \times n}$ *have rank* $r \leq \min\{d, n\}$, *and let* $\mathbf{P}$ *be defined by* (15). *Then,*

$$\mathrm{cond}(\mathbf{P}\,\boldsymbol{\Phi}) = \mathrm{cond}(\boldsymbol{\Phi}) \cdot \frac{\sigma_{\min}^2(\boldsymbol{\Phi}) + \gamma}{\sigma_{\max}^2(\boldsymbol{\Phi}) + \gamma} \leq \mathrm{cond}(\boldsymbol{\Phi}),$$

*where* $\sigma_{\max}(\boldsymbol{\Phi})$ *and* $\sigma_{\min}(\boldsymbol{\Phi})$ *are the largest and smallest singular values of* $\boldsymbol{\Phi}$, *respectively.*

This lemma shows that left-multiplying $\boldsymbol{\Phi}$ by $(\boldsymbol{\Phi}\boldsymbol{\Phi}^{\top} + \gamma\,\mathbf{I})^{-1}$ compresses its singular values, reducing the condition number unless $\boldsymbol{\Phi}$ originally had all singular values equal (Appendix B). Under the assumption that $\boldsymbol{\Phi}$ has $d$ nonzero eigenvalues (i.e., $\boldsymbol{\Phi}$ is full rank), any $\gamma > 0$ strictly improves $\mathrm{cond}(\boldsymbol{\Phi})$. Whether this improvement is sufficient to satisfy the RIP condition (with small $\delta$) depends on the chosen $\gamma$.

We now define a VAE architecture that simultaneously seeks to evaluate the optimal sparse coefficients $\mathbf{z}$ and the regularization term $\gamma$, to target satisfying $(\kappa, \delta)$-RIP for some $\delta < 1$. The key idea is to *precondition* $\boldsymbol{\Phi}$ by $\mathbf{P}$ directly within the VAE training process, potentially revealing a global optimum that meets the RIP condition. Specifically, the decoder's Gaussian distribution is updated to:

$$p_\theta(\mathbf{x} \mid \mathbf{z}) = \mathcal{N}\big(\mathbf{x}; \mathbf{P}\boldsymbol{\Phi}\,\mathrm{diag}(\mathbf{w}_x)\,\mathbf{z}, \gamma\,\mathbf{I}\big), \quad (16)$$

where we reuse the same $\gamma$ for both the decoder covariance $\gamma\,\mathbf{I}$ and the preconditioning term $\mathbf{P}$. This unifies the effect of $\gamma$ on preconditioning and the variance for the sparse components in the decoder. As $\boldsymbol{\Phi}$ is of full rank, its condition number improves through multiplication by $\mathbf{P}$. Similarly, we adjust the mean vector $\boldsymbol{\mu}_z = \mathbf{W}_z\mathbf{P}\,\mathbf{x}$ in the encoder and maintain the full-rank covariance $\mathbf{S}\mathbf{S}^{\top}$. Consequently, the original VAE loss in (13) becomes:

$$\mathcal{L}(\theta, \phi) = \mathbb{E}_{q_\phi(\mathbf{z}|\mathbf{x})}\left[\frac{1}{\gamma}\left\|\mathbf{P}\mathbf{x} - \mathbf{P}\boldsymbol{\Phi}\,\mathrm{diag}(\mathbf{w}_x)\,\mathbf{z}\right\|^2\right]$$
$$+ d\log\gamma + \mathrm{tr}\big[\mathbf{S}\,\mathbf{S}^{\top}\big]$$
$$- \log\big|\mathbf{S}\,\mathbf{S}^{\top}\big| + \|\mathbf{W}_z\,\mathbf{P}\,\mathbf{x}\|_2^2, \quad (17)$$

and we show that (17) can eliminate unwanted local minima if the optimal $\gamma$ also yields an $\mathbf{P}^*\boldsymbol{\Phi}$ satisfying $(\kappa, \delta)$-RIP for some $\delta < 1$.

**Theorem 4.4** (VAE-Induced Preconditioning). *Any minimum* $\{\hat{\mathbf{w}}_x, \hat{\mathbf{W}}_z, \hat{\mathbf{S}}\}$ *of* (17) *for a given* $\gamma^*$ *is a global minimum if,* $\hat{\mathbf{P}}\boldsymbol{\Phi} = \big(\boldsymbol{\Phi}\,\boldsymbol{\Phi}^{\top} + \gamma^*\,\mathbf{I}\big)^{-1}\boldsymbol{\Phi}$ *satisfies the RIP condition with* $\delta < 1$. *Under this circumstance, the global minimum matches the optimal sparse solution for SLR in* (6) *only if* $\gamma^*$ *is small enough to meet the limiting conditions of Theorem 4.2. The resulting sparse coefficient* $\hat{\mathbf{z}}$ *is given by:*

$$\mathrm{diag}[\hat{\mathbf{w}}_x]^2\,\boldsymbol{\Phi}^{\top}\hat{\mathbf{P}}^{\top}\left(\hat{\mathbf{P}}\boldsymbol{\Phi}\,\mathrm{diag}[\hat{\mathbf{w}}_x]^2\,\boldsymbol{\Phi}^{\top}\hat{\mathbf{P}}^{\top}\right)^{-1}\hat{\mathbf{P}}\mathbf{x}. \quad (18)$$

The $\gamma$ term is a training parameter in Theorem 4.4, which effectively penalizes the condition number of $\boldsymbol{\Phi}$ in the VAE objective (17). As shown in Lemma 4.3, any positive value of $\gamma$ improves the condition number of $\mathbf{P}\boldsymbol{\Phi}$ compared to $\boldsymbol{\Phi}$. Therefore, adding $\gamma$ to the overall loss function improves the effective condition number of $\boldsymbol{\Phi}$, pushing it to satisfy the RIP property (Appendix C). This assists in achieving the optimal sparse solutions for ill-conditioned design matrices where LASSO fails (Kelner et al., 2022a).

**Interpretation of Theorem 4.4:** From the proof of Theorem 4.4, we see that there can be multiple values of $\gamma^*$ that make the preconditioned design matrix $\tilde{\boldsymbol{\Phi}}$ satisfy the RIP condition. For each such $\gamma^*$, the VAE smoothens out bad local minima and converges to global minima. However, Theorem 4.2 tells us that the final solution only matches the true sparse representation when we choose $\gamma$ approaching zero. Thus, the desired solution for $\gamma^*$ must be sufficiently small so that the global minimum of (17) aligns with the limiting case of $\gamma \to 0$ (using the preconditioned $\boldsymbol{\Phi}$ and $\mathbf{x}$). Consequently, if an ill-conditioned design matrix can be preconditioned with a suitably small $\gamma^*$ to satisfy the RIP condition, a VAE can solve the corresponding SLR problem. Although this requirement is restrictive, it does occur in practice for certain ill-conditioned design matrices, as shown in Section 5.

### 4.2. Additional Perspectives

We now highlight some of the additional perspectives that can be drawn from the results in this section. They include a comparison with statistical methods for SLR, such as sparse Bayesian learning (SBL), and the role of VAEs in addressing challenging cases of ill-conditioned SLR design matrices.

#### 4.2.1. COMPARISON WITH SPARSE BAYESIAN LEARNING

Addressing one of the key discussion points brought forth by an ICML 2025 reviewer during the evaluation of this paper, we compare our approach with Sparse Bayesian learning (SBL) (Tipping, 2001; Wipf et al., 2011; 2015)

in detail. SBL is an empirical Bayesian method for solving SLR, which outperforms LASSO in support recovery accuracy (Lin et al., 2022). However, it relies on type II maximum likelihood estimation, which benefits from prior knowledge of the distribution of the nonzero coefficients in **z**. In the absence of this prior information, SBL resorts to expectation maximization (EM) algorithms to optimize the SLR objective. Our proposed VAE addresses the limitations of SBL as described below:

1. **Global Optimum**: The global minimum in SBL for SLR corresponds to the optimally sparse coefficients (Wipf & Rao, 2004); however, EM algorithms may converge to spurious local minima. Our VAE architecture smooths the loss landscape, enabling convergence to the global minimum, which coincides with the optimal sparse solution (Theorem 4.4).

2. **Handling Ill-conditioned Matrices**: Design matrices that violate the RIP bound can cause numerical instability in the matrix inversion step of SBL's EM algorithm, reducing the sparse recovery rate. In contrast, our VAE preconditions the design matrices to satisfy the RIP bound, achieving a higher recovery rate at the same sparsity level.

3. **Computational Complexity**: Each EM iteration involves a matrix inversion of time complexity $O(n^3)$, making SBL computationally expensive and limiting its scalability. In contrast, training our linear VAE via backpropagation runs in $O(n^2)$ time per epoch, since it avoids matrix inversion.

4. **Implicit Sparsity Prior**: Our VAE overcomes the lack of prior knowledge on the sparse coefficients by incorporating a trainable diagonal matrix $\text{diag}(\mathbf{w})$ in the decoder, which implicitly captures sparsity information without requiring an explicit prior distribution.

#### 4.2.2. HARD CASES OF ILL-CONDITIONED DESIGN MATRICES

While Theorem 4.4 shows that VAEs can narrow the eigenvalue range of an ill-conditioned design matrix, it remains to be seen whether different kinds of ill-conditioned matrices truly benefit from such preconditioning and can be considered as a future direction. Interestingly, there are certain special classes of ill-conditioned matrices with a unique sparse solution, for which achieving lower error than LASSO is tantamount to breaking post-quantum assumptions in lattice problems (Gupte et al., 2024). As a result, some matrices may stay intractable even if a VAE-based approach reduces their eigenvalue separation. Nonetheless, whenever preconditioning succeeds in shrinking the spread of eigenvalues, the improved conditioning has the potential

to facilitate sparse recovery. However, the impact of preconditioning on the solvability of SLR depends on the matrix structure and how significantly its eigenvalues can be equalized. Hereafter, to validate our theoretical results in this section, we conduct the corresponding empirical analysis of SLR using various design matrices in Section 5.

## 5. Empirical Validation

To re-iterate, our VAE-based sparse recovery aims to identify the positions of non-zero coefficients rather than directly estimating their values. Once the correct support (non-zero coefficient positions) is identified, the SLR problem simplifies significantly and can be solved using ordinary least squares (OLS). Under a full-rank design matrix and Gauss-Markov assumptions, OLS is an unbiased estimator leading to optimal sparse recovery with no bias.

Our experiments cover three types of design matrices in a noiseless setting [1]. We compare our approach, presented in Theorems 4.2 and 4.4, with established methods like LASSO (Tibshirani, 1996), SBL (Lin et al., 2022), and augmented basis pursuit (Kelner et al., 2024), for both well-conditioned and ill-conditioned design matrices. The VAE is trained using SGD with **x**. We first select $\kappa$ uniformly random sparse locations within an $n$-dimensional vector, then sample $\kappa$ non-zero coefficients from a standard normal distribution to obtain sparse coefficients **z**. Thereafter, **x** is generated by multiplying **z** with the design matrix $\mathbf{\Phi}$. These **x** values are used to train the VAE using SGD.

All the relevant codes and a detailed user manual for replicating the experiments in this work are available at `https://github.com/SEAL-IIT-KGP/Be-a-Goldfish-Solving-SLR-using-VAE`.

### 5.1. Design Matrix from a Standard Gaussian Distribution

In our first scenario, we construct a design matrix $\mathbf{\Phi}$ by drawing its features from a standard Gaussian distribution ($\mathbf{\Phi}_{i,:} \sim \mathcal{N}(0, \mathbf{I})$). We set $n = 200$ features and $d = 100$ observations, then recover the sparse coefficients $\hat{\mathbf{z}}$ at different sparsity levels $\kappa \in \{2, 10, 20, 30, 40, 50, 60\}$. Repeating the experiment 10 times with randomly sampled features and non-zero coefficients yields the sparse support recovery rates shown in Fig. 1(a). Compared to standard LASSO and SBL, our linear VAE approach, as presented in Theorem 4.2, achieves a higher recovery rate, particularly at lower sparsity.

For LASSO, the probability of recovering the correct sparse coefficients is governed by a control parameter $\theta_c = \frac{n - \kappa - 1}{2\,\kappa\,\log(d - \kappa)}$, when the features are drawn from a stan-

---

[1]Our technique also applies to noisy sparse recovery, as discussed in Appendix D

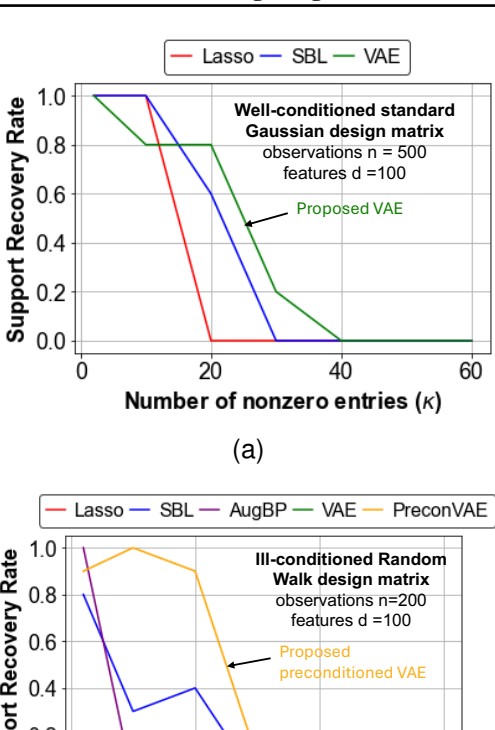

(a)

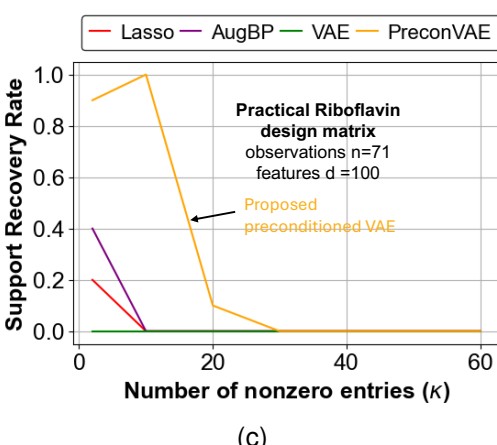

(b)

(c)

*Figure 1.* Sparse support recovery rate for SLR vs. increasing number of non-zero entries in $\mathbf{z}$ or the $\kappa$-sparsity level for features of design matrix $\mathbf{\Phi}_{i,:}$ sampled from (a) standard Gaussian, (b) Gaussian random walk, and (c) Riboflavin dataset (Bühlmann et al., 2014) for different recovery methods.

dard Gaussian (Wainwright, 2006). This sample complexity relationship aligns with our observations in Fig. 1(a), where LASSO's recovery rate diminishes as $\kappa$ increases. Indeed, for $\kappa > 30$, LASSO's recovery rate falls to zero, reflecting the near-zero success probability reported in (Wainwright, 2006) for $\theta_c < 0.22$. By contrast, our VAE-based method

exhibits a higher tolerance to lower sparsity (i.e., larger $\kappa$). We conjecture that this robustness arises because the VAE can learn both the mean and covariance parameters of the sparse coefficients. In contrast, LASSO focuses explicitly on reducing its $\ell_1$ norm for promoting sparsity.

### 5.2. Design Matrix from a Gaussian Random Walk

Next, we construct an ill-conditioned $\mathbf{\Phi}$ by drawing its features from a Gaussian random walk distribution. Specifically, the rows of $\mathbf{\Phi}$ are i.i.d. copies of the elements of a random walk with $\mathbf{\Phi}_{i,:} = \{r_1, r_2, \ldots, r_k\}$ where,

$$r_i = r_{i-1} + z \,\forall\, i > 1,$$
$$z \sim \mathcal{N}(0,1), \quad r_1 \sim \mathcal{N}(0,1). \tag{19}$$

This construction yields a covariance matrix $\mathbf{\Sigma}_{i,j} = \min(i, j)$, which has a high condition number. As shown in (Kelner et al., 2022b), such a design matrix cannot be solved using LASSO without appropriate preconditioning. We employ the same matrix dimension and the sparsity values as the previous case of standard Gaussian covariance in Section 5.1.

Our empirical results in Fig. 1(b) confirm that, while solutions without preconditioning fail, SBL (Lin et al., 2022), augmented basis pursuit (Kelner et al., 2024), and our preconditioned VAE (Theorem 4.4) recover sparse solutions for low sparsity levels (e.g., $\kappa = 2$). However, as $\kappa$ increases (i.e., as sparsity decreases), the preconditioned VAE outperforms other methods by maintaining a higher recovery rate until $\kappa \le 20$. We conjecture that the VAE's learned preconditioning factor $\gamma$ effectively improves the condition number of $\mathbf{\Phi}$ and enforces the RIP condition for small $\delta$, leading to high recovery rates for $\kappa < 20$.

### 5.3. Practical Design Matrix from Riboflavin Dataset

Lastly, we evaluate SLR on a design matrix constructed from real-life biological measurements in the riboflavin dataset (Bühlmann et al., 2014). The genetic features in this dataset are correlated, making $\mathbf{\Phi}$ inherently ill-conditioned. Practical constraints limit the number of features to $d = 100$ and the number of observations to $n = 71$. Given the sample complexity for successful LASSO is in the order of $\Omega(\kappa \log(d))$ (Kelner et al., 2022b), the smaller sample size in this case tests the low-sparsity tolerance of our preconditioned VAE (Theorem 4.4) in a particularly challenging setting. Nevertheless, Fig. 1(c) shows that the preconditioned VAE achieves the highest recovery rate for $\kappa \le 10$. Although $\kappa = 10$ represents a higher sparsity limit than in Fig. 1(a) and (b), our method still surpasses existing alternatives under these challenging conditions. Increasing the number of features to $d = 200$ leads to similar support recovery as Fig. 1(c), as shown in additional experimental results in Appendix D.

### 5.4. Impact of SLR Parameters on Sparse Recovery

As suggested by reviewers during the ICML 2025 rebuttal phase, we evaluate the effect of changing SLR parameters on the support recovery performance using our proposed VAE. All detailed results can be found in Appendix D. We summarize the key takeaways as follows:

1. **Design matrix dimensions:** The preconditioned VAE consistently outperforms competing methods in support recovery rate, both as the number of features $n$ and the number of observations $d$ increase. The reason is the presence of more information for solving the SLR problem.

2. **Nonzero coefficient distribution:** Our VAE's focus on support identification makes it insensitive to the nonzero coefficient distribution. Empirical results confirm similar recovery rates across various distributions of the nonzero coefficients.

3. **Noise level:** To assess the impact of additive noise $\eta$, we conducted experiments at different SNR levels. While all methods improve with higher SNR, our VAE achieves superior recovery rates even at lower SNR. This robustness can be further enhanced by preprocessing techniques such as filtering or mixture-of-Gaussians models (Guo et al., 2021).

### 5.5. Fixed vs. Trainable $\gamma$ during Preconditioning

Theorems 4.2 and 4.4 show the existence of optimal sparse solutions when $\gamma \to 0$, and our empirical results in achieve the same for most cases except when SGD fails to attain the limiting solution. The trainable $\gamma$ assists in achieving a higher support recovery rate of the preconditioned VAE for ill-conditioned SLR compared to fixed $\gamma$. Although proposed VAE architecture can achieve no bad local minima condition for a fixed $\gamma$, optimal sparse recovery with $\gamma \to 0$ is contingent on the success of the optimization algorithm. Imperfect optimization can hinder achieving this ideal scenario, as evidenced by our empirical results in Appendix D.

### 5.6. Insights on Higher Tolerance to Low-Sparsity

A higher tolerance for low sparsity translates into obtaining an optimal sparse solution using fewer observations, thereby reducing data collection overhead. This advantage is particularly beneficial in biological contexts, where datasets can contain millions of features (Rives et al., 2021). To encourage the adoption of VAE-based methods in such settings, a solid theoretical understanding of SLR's solvability is essential. Although prior work (Kelner et al., 2020; 2024; Wainwright, 2006) has explored sample complexity limits for LASSO-based SLR, our findings suggest that variational methods, such as VAEs, may offer greater flexibility in handling lower sparsity. With all the above interesting insights, next, we conclude our paper.

## 6. Conclusion and Future Work

In this work, we broaden our understanding of the local minima smoothing property of VAEs in the context of a well-known NP-hard problem in high-dimensional statistics: Sparse Linear Regression (SLR). Our primary focus is on scenarios involving ill-conditioned design matrices and low sparsity, where classic methods such as LASSO often fail. A key limitation of LASSO is that it reformulates the $\ell_0$ sparsity constraint into an $\ell_1$ regularization objective, which can struggle to recover the optimal solution when the design matrix is poorly conditioned or the sparsity level is low.

By contrast, our central insight is that VAEs can simultaneously impose sparsity constraints and learn the underlying distribution of sparse coefficients, enabling more informed feature selection. Leveraging this capability, we propose a VAE architecture that intrinsically preconditions ill-conditioned design matrices, thereby surpassing LASSO in specific matrix classes. Across different types of design matrices, the VAE-based approach consistently demonstrates a higher tolerance for sparsity compared to LASSO and previously introduced preconditioning techniques. Overall, our findings expand the applicability of VAEs to NP-hard sparse inverse problems—an area where generative models have yet to be thoroughly explored. This work opens several promising research directions:

1. **SLR in Challenging Domains:** The ability of VAEs to handle low-sparsity and ill-conditioned design matrices in SLR is highly relevant to applications such as feature selection for privacy-preserving machine learning (Akavia et al., 2024; Li et al., 2021), neural imaging of brain function (Shen et al., 2022), and genome selection in cancer research (Fan et al., 2024).

2. **Theoretical Underpinnings for VAE-Based Solutions:** Our results suggest that existing limits on the solvability of SLR under LASSO can be pushed by employing variational methods for sparse recovery, which account for both the sparsity constraint and the distribution of sparse coefficients. Future work includes investigating new preconditioning strategies for ill-conditioned design matrices and sample complexity bounds for accurate recovery in low-sparsity regimes.

By uniting insights from generative modeling and high-dimensional statistics, our work broadens the theoretically grounded approach of VAEs in solving NP-hard inverse problems under specific constraints.

## Impact Statement

This work bridges the fields of generative modeling and high-dimensional statistics. We leverage VAEs to solve the well-known NP-hard problem of sparse linear regression (SLR). Traditional SLR methods often fail in real-world settings that feature correlated features or have a limited number of observations. VAEs can intrinsically "precondition" these matrices under certain conditions, leading to a higher recovery rate.

From a societal perspective, this result holds significant promise. It enables the accurate recovery of sparse signals in domains such as neurological imaging, cancer genomics, and system identification. This VAE-based approach provides faster and more precise insights, which can improve healthcare outcomes and drive scientific progress. Additionally, in privacy-preserving machine learning, a reliable SLR method can reduce training data requirements. This, in turn, helps safeguard sensitive information while lowering communication costs. Looking forward, employing VAEs for NP-hard inverse problems opens new avenues for innovation in machine learning and statistical modeling. Researchers and practitioners can harness these generative capabilities to tackle applications where data is limited. Therefore, our proposed approach makes a meaningful contribution to broader scientific and societal benefits.

**Acknowledgement:** Our profound appreciation goes to the anonymous reviewers for their constructive feedback, which greatly refined this paper. Additionally, the authors are thankful for the partial funding received from the Center for Hardware-Security Entrepreneurship Research & Development (C-HERD) and Information Security Education and Awareness (ISEA), both initiatives of MeitY, Govt. of India. Lastly, the authors were also partially supported by Qualcomm with Reference under Grant IND-492686.

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

# Appendix

In this section, we provide the proofs for the lemmas and theorems used in this paper. We begin by listing the notation for both variables and functions. Boldface uppercase Greek symbols (e.g., $\boldsymbol{\Phi}$) and boldface uppercase Latin letters (e.g., $\mathbf{P}$) denote matrices. Vectors are denoted by boldface lowercase letters (e.g., $\boldsymbol{\mu}_x$ or $\mathbf{x}$). All scalars appear in regular font weight and lowercase letters. To minimize notation overload, we occasionally reuse symbols for related variables. For example, $\mathbf{x}$ represents both the input vector to the VAE and the observation vector in SLR.

*Table 1.* List of all variables used throughout the paper

| Variable | Dimension | Description | Variable | Dimension | Description |
|---|---|---|---|---|---|
| | | Sparse Linear Regression (SLR) | | | |
| $d$ | Scalar | Observation dimension | $n$ | Scalar | Coefficient dimension |
| $\mathbf{x}$ | $\mathbb{R}^d$ | Observation vector | $\mathbf{z}$ | $\mathbb{R}^n$ | Coefficient vector |
| $\boldsymbol{\Phi}$ | $\mathbb{R}^{d \times n}$ | Design matrix | $\eta$ | $\mathbb{R}^d$ | Noise vector |
| $\boldsymbol{\Phi}_{i,:}$ | $\mathbb{R}^n$ | $i^{th}$ row of $\boldsymbol{\Phi}$ | $\boldsymbol{\Phi}_{:,j}$ | $\mathbb{R}^d$ | $j^{th}$ column of $\boldsymbol{\Phi}$ |
| $\mathbf{P}$ | $\mathbb{R}^{d \times d}$ | Preconditioning matrix | $\tilde{\eta}$ | $\mathbb{R}^d$ | Preconditioned $\eta$ |
| $\tilde{\boldsymbol{\Phi}}$ | $\mathbb{R}^{d \times n}$ | Preconditioned $\boldsymbol{\Phi}$ | $\tilde{\mathbf{x}}$ | $\mathbb{R}^d$ | Preconditioned $\mathbf{x}$ |
| $\delta$ | Scalar | Restricted isometry constant of $\boldsymbol{\Phi}$ | $\boldsymbol{\Sigma}$ | $\mathbb{R}^{n \times n}$ | Gram/ Covariance matrix of $\boldsymbol{\Phi}^{\dagger}$ |
| $\lambda$ | Scalar | Regularizer for LASSO | $\sigma$ | Scalar | Eigenvalues of a Matrix |
| $\kappa$ | Scalar | Sparsity value | $\mathbf{z}^*$ | $\mathbb{R}^n$ | Ground truth $\mathbf{z}$ |
| | | Variational Autoencoder (VAE) | | | |
| $d$ | Scalar | Input dimension | $n$ | Scalar | Latent Dimension |
| $\mathbf{x}$ | $\mathbb{R}^d$ | Input vector | $\mathbf{z}$ | $\mathbb{R}^n$ | Latent Vector |
| $\gamma$ | Scalar | Decoder variance | $\mathbf{W}_x$ | $\mathbb{R}^{d \times n}$ | Linear decoder matrix |
| $\mathbf{w}_x$ or $\mathbf{s}$ | $\mathbb{R}^n$ | Decoder parameter | $\mathbf{w}$ | $\mathbb{R}^n$ | Element-wise squared $\mathbf{w}_x{}^2$ |
| $\boldsymbol{\mu}_x$ | $\mathbb{R}^d$ | Decoder mean | $\boldsymbol{\Sigma}_x$ | $\mathbb{R}^{d \times d}$ | Decoder covariance $\gamma \mathbf{I}$ |
| $\mathbf{b}_x$ | $\mathbb{R}^d$ | Linear decoder bias | $\mathbf{b}_z$ | $\mathbb{R}^n$ | Linear encoder bias |
| $\boldsymbol{\mu}_z$ | $\mathbb{R}^n$ | Encoder mean | $\boldsymbol{\Sigma}_z$ | $\mathbb{R}^{n \times n}$ | Encoder covariance |
| $\mathbf{I}$ | $\mathbb{R}^{d \times d}$ or $\mathbb{R}^{n \times n}$ | Identity matrix | $(\theta, \phi)$ | Arbitrary | (Encoder, Decoder) parameter |

$\dagger$ We also use $\boldsymbol{\Sigma}$ for denoting the singular values matrix in singular value decomposition (SVD).

*Table 2.* List of all functions used throughout the paper

| Function | Description |
|---|---|
| | Sparse Linear Regression (SLR) |
| $\|\|\mathbf{z}\|\|_p$ | $\ell_p$ norm of $\mathbf{z}$ where $p \in \{0, 1, 2\}$ |
| $\text{cond}(\boldsymbol{\Sigma})$ | Condition number for $\boldsymbol{\Sigma}$ |
| | Variational Autoencoder (VAE) |
| $\mathcal{N}(\mathbf{x}\|\boldsymbol{\mu}, \boldsymbol{\Sigma})$ | Multivariate Gaussian distribution over $\mathbf{x}$ with mean $\boldsymbol{\mu}$ and Covariance $\boldsymbol{\Sigma}$ |
| $p_\theta(\mathbf{x})$ | Marginal probability distribution of $\mathbf{x}$ parameterized by $\theta$ |
| $p_\theta(\mathbf{x}, \mathbf{z})$ | Joint probability distribution of $\mathbf{x}, \mathbf{z}$ parameterized by $\theta$ |
| $p_\theta(\mathbf{x}\|\mathbf{z})$ | Conditional likelihood distribution of $\mathbf{x}$ given $\mathbf{z}$ parameterized by $\theta$ |
| $p(\mathbf{z})$ | Prior distribution of the latent variable $\mathbf{z}$ |
| $p_\theta(\mathbf{z}\|\mathbf{x})$ | Posterior probability distribution of $\mathbf{z}$ given $\mathbf{x}$ parameterized by $\theta$ |
| $q_\phi(\mathbf{z}\|\mathbf{x})$ | Approximate posterior distribution of $\mathbf{z}$ given $\mathbf{x}$ parameterized by $\phi$ |
| $\mathbb{E}_{q_\phi(\mathbf{z}\|\mathbf{x})}(\mathbf{x})$ | Expectation of $\mathbf{x}$ over the probability distribution $q_\phi(\mathbf{z}\|\mathbf{x})$ |
| $\text{KL}[q_\phi(\mathbf{z}\|\mathbf{x})\|\|p(z)]$ | Kullback-Liebler divergence between two distributions $q_\phi(\mathbf{z}\|\mathbf{x}), p(z)$ |
| $\mathcal{L}(\theta, \phi)$ | VAE loss function w.r.t. encoder $\phi$ and decoder $\theta$ parameters |

Furthermore, we denote local or global minima with a "hat," for instance, $\hat{\mathbf{w}}_x$ represents the vector $\mathbf{w}_x$ that minimizes the VAE loss $\mathcal{L}(\theta, \phi)$. We use "star," for the ground-truth variable $\mathbf{z}^*$ in (4) and for $\gamma^*$ in Theorem 4.4. Preconditioned matrices and vectors carry a "tilde," for example, $\tilde{\boldsymbol{\Phi}} = \mathbf{P}\boldsymbol{\Phi}$ for the preconditioned design matrix.

## A. Proof of Theorem 4.2

We first optimize over the encoder parameters to obtain a condensed loss, which is then analyzed with respect to the decoder parameter, the latter occupying the majority of the proof. For the encoder-decoder architecture choice objective in (13) reduces to:

$$
\mathcal{L}(\theta, \phi) = \frac{1}{\gamma} \|\mathbf{x} - \mathbf{\Phi} \operatorname{diag}[\mathbf{w}_x] \mathbf{W}_z \mathbf{x}\|_2^2 + \frac{1}{\gamma} \operatorname{tr}\left[\operatorname{diag}[\mathbf{w}_x] \mathbf{\Phi}^\top \mathbf{\Phi} \operatorname{diag}[\mathbf{w}_x] \mathbf{S}\,\mathbf{S}^\top\right]
$$
$$
+ d \,\log \gamma + \operatorname{tr}\left[\mathbf{S}\,\mathbf{S}^\top\right] - \log\left|\mathbf{S}\,\mathbf{S}^\top\right| + \|\mathbf{W}_z \mathbf{x}\|_2^2, \tag{20}
$$

with $\theta = \{\mathbf{w}_x, \gamma\}$ and $\phi = \{\mathbf{W}_z, \mathbf{S}\}$. Although this loss is nonconvex, we can still take derivatives with respect to $\mathbf{S}\,\mathbf{S}^\top$ to show the existence of a single stationary point. In doing so, we find that

$$
\mathbf{S}\,\mathbf{S}^\top = \left(\frac{1}{\gamma} \operatorname{diag}[\mathbf{w}_x] \mathbf{\Phi}^\top \mathbf{\Phi} \operatorname{diag}[\mathbf{w}_x] + \mathbf{I}\right)^{-1} \tag{21}
$$

is the unique minimizer. Note that the identity matrix in (21) is of dimension $n \times n$. Substituting (21) into (20), yields the revised cost as:

$$
\mathcal{L}(\theta, \phi) = \frac{1}{\gamma} \|\mathbf{x} - \mathbf{\Phi} \operatorname{diag}[\mathbf{w}_x] \mathbf{W}_z \mathbf{x}\|_2^2 + d \,\log \gamma + \log\left|\frac{1}{\gamma} \operatorname{diag}[\mathbf{w}_x] \mathbf{\Phi}^\top \mathbf{\Phi} \operatorname{diag}[\mathbf{w}_x] + \mathbf{I}\right| + \|\mathbf{W}_z \mathbf{x}\|_2^2. \tag{22}
$$

Since (22) is convex in $\mathbf{W}_z$, we can also optimize these parameters without encountering local minima issues, noting that the optimal value satisfies:

$$
\mathbf{W}_z \mathbf{x} = \operatorname{diag}[\mathbf{w}_x] \mathbf{\Phi}^\top \left(\mathbf{\Phi} \operatorname{diag}[\mathbf{w}_x]^2 \mathbf{\Phi}^\top + \gamma\,\mathbf{I}\right)^{-1} \mathbf{x}. \tag{23}
$$

Note that the $\mathbf{I}$ in (23) is of dimension $d \times d$, which we obtain after applying standard determinant identities (e.g., the Woodbury matrix identity). Column-wise, this expression is tantamount to $\boldsymbol{\mu}_x(\mathbf{x}; \phi) = \mathbf{W}_z \mathbf{x}$. To simplify notation, let us write $\mathbf{w} \triangleq \mathbf{w}_x^2 \geq 0$ (elementwise). We also define $\mathbf{W} \triangleq \operatorname{diag}[\mathbf{w}]$. Substituting (23) into (22) further reduces the remaining parameters. The VAE loss can be equivalently expressed as:

$$
\mathcal{L}(\mathbf{w}, \gamma) = \mathbf{x}^\top \boldsymbol{\Sigma}_x^{-1} \mathbf{x} + \log|\boldsymbol{\Sigma}_x|, \quad \text{with } \boldsymbol{\Sigma}_x \triangleq \mathbf{\Phi}\,\mathbf{W}\,\mathbf{\Phi}^\top + \gamma\,\mathbf{I}. \tag{24}
$$

We now show that for a fixed $\gamma$ any minimum of (24) is a global minimum and that, in the limiting case $\gamma \to 0$, the global minimum approaches the optimal sparse representation for SLR, provided the design matrix is well-conditioned satisfying the RIP condition with a small delta.

### A.1 Deriving the Stationarity Conditions

Let us denote $\mathbf{\Phi} \operatorname{diag}[\mathbf{w}] \mathbf{\Phi}^\top + \gamma\,\mathbf{I}$ using the matrix $\boldsymbol{\Sigma}(\mathbf{w})$. The VAE loss then depends only on $\mathbf{w}$ and is given by:

$$
\mathcal{L}(\mathbf{w}) = \mathbf{x}^\top \boldsymbol{\Sigma}^{-1}(\mathbf{w})\,\mathbf{x} + \log|\boldsymbol{\Sigma}(\mathbf{w})|. \tag{25}
$$

We now compute the stationary points by finding $\dfrac{\partial \mathcal{L}}{\partial w_j}$ for each $w_j \in \mathbf{w}$, for each of the two terms in (25).

**Derivative of the Inverse-Quadratic Term.**

Consider $T_1(\mathbf{w}) = \mathbf{x}^\top \boldsymbol{\Sigma}^{-1}(\mathbf{w})\,\mathbf{x}$. When the following matrix-calculus identity for an arbitrary matrix $\mathbf{A}$:

$$
\frac{\partial}{\partial \theta}\left[\mathbf{A}(\theta)^{-1}\right] = -\mathbf{A}(\theta)^{-1} \frac{\partial \mathbf{A}(\theta)}{\partial \theta} \mathbf{A}(\theta)^{-1}, \tag{26}
$$

is applied to $\boldsymbol{\Sigma}^{-1}(\mathbf{w})$ we get:

$$
\frac{\partial}{\partial w_j} \boldsymbol{\Sigma}^{-1}(\mathbf{w}) = -\boldsymbol{\Sigma}^{-1}(\mathbf{w}) \frac{\partial \boldsymbol{\Sigma}(\mathbf{w})}{\partial w_j} \boldsymbol{\Sigma}^{-1}(\mathbf{w}). \tag{27}
$$

Since $\dfrac{\partial \boldsymbol{\Sigma}(\mathbf{w})}{\partial w_j} = \boldsymbol{\phi}_j\,\boldsymbol{\phi}_j^\top$ (because differentiating $\mathrm{diag}[\mathbf{w}]$ with respect to $w_j$ picks out the $j$th diagonal element, yielding the $j$th rank-1 component), we substitute it back into (27) to get:

$$\frac{\partial}{\partial w_j}\,\boldsymbol{\Sigma}^{-1}(\mathbf{w}) \;=\; -\,\boldsymbol{\Sigma}^{-1}(\mathbf{w})\,\big(\boldsymbol{\phi}_j\boldsymbol{\phi}_j^\top\big)\,\boldsymbol{\Sigma}^{-1}(\mathbf{w}). \tag{28}$$

By the chain rule of differentiation,

$$\frac{\partial}{\partial w_j}\Big(\mathbf{x}^\top\boldsymbol{\Sigma}^{-1}(\mathbf{w})\,\mathbf{x}\Big) \;=\; \mathbf{x}^\top\Big[\frac{\partial \boldsymbol{\Sigma}^{-1}(\mathbf{w})}{\partial w_j}\Big]\mathbf{x} \;=\; \mathbf{x}^\top\Big[-\boldsymbol{\Sigma}^{-1}(\mathbf{w})\,\boldsymbol{\phi}_j\boldsymbol{\phi}_j^\top\,\boldsymbol{\Sigma}^{-1}(\mathbf{w})\Big]\mathbf{x}. \tag{29}$$

Since $\mathbf{x}^\top \mathbf{A}\,\mathbf{x}$ is a scalar for an arbitrary matrix $\mathbf{A}$, we can rewrite (29) as:

$$\frac{\partial}{\partial w_j}\Big(\mathbf{x}^\top\boldsymbol{\Sigma}^{-1}(\mathbf{w})\,\mathbf{x}\Big) \;=\; -\,\big(\boldsymbol{\phi}_j^\top\boldsymbol{\Sigma}^{-1}(\mathbf{w})\,\mathbf{x}\big)^2. \tag{30}$$

**Derivative of the Log-Det Term.**

Next, consider the second term, $T_2(\mathbf{w}) \;=\; \log\big|\boldsymbol{\Sigma}(\mathbf{w})\big|$. Applying the identity:

$$\frac{\partial}{\partial \theta}\,\log\big|\mathbf{A}(\theta)\big| \;=\; \mathrm{trace}\Big[\mathbf{A}(\theta)^{-1}\,\tfrac{\partial \mathbf{A}(\theta)}{\partial \theta}\Big], \tag{31}$$

we get,

$$\frac{\partial}{\partial w_j}\,\log\big|\boldsymbol{\Sigma}(\mathbf{w})\big| \;=\; \mathrm{trace}\Big[\boldsymbol{\Sigma}^{-1}(\mathbf{w})\,\tfrac{\partial \boldsymbol{\Sigma}(\mathbf{w})}{\partial w_j}\Big] \;=\; \mathrm{trace}\Big[\boldsymbol{\Sigma}^{-1}(\mathbf{w})\,\boldsymbol{\phi}_j\boldsymbol{\phi}_j^\top\Big]. \tag{32}$$

Using the cyclic property of trace, $\mathrm{trace}(\mathbf{A}\,\mathbf{B}) = \mathrm{trace}(\mathbf{B}\,\mathbf{A})$, and the fact that $\mathrm{trace}(\mathbf{u}\,\mathbf{v}^\top) = \mathbf{v}^\top\mathbf{u}$ for vectors, we get

$$\frac{\partial}{\partial w_j}\,\log\big|\boldsymbol{\Sigma}(\mathbf{w})\big| \;=\; \boldsymbol{\phi}_j^\top\boldsymbol{\Sigma}^{-1}(\mathbf{w})\,\boldsymbol{\phi}_j. \tag{33}$$

Adding the two contributions and setting the derivative w.r.t. $w_j$ to zero,

$$\frac{\partial \mathcal{L}(\mathbf{w})}{\partial w_j} \;=\; -\big(\boldsymbol{\phi}_j^\top\boldsymbol{\Sigma}^{-1}(\mathbf{w})\,\mathbf{x}\big)^2 \;+\; \boldsymbol{\phi}_j^\top\boldsymbol{\Sigma}^{-1}(\mathbf{w})\,\boldsymbol{\phi}_j \;=\; 0 \tag{34}$$

$$\implies \big(\boldsymbol{\phi}_j^\top\boldsymbol{\Sigma}^{-1}(\mathbf{w})\,\mathbf{x}\big)^2 \;=\; \boldsymbol{\phi}_j^\top\boldsymbol{\Sigma}^{-1}(\mathbf{w})\,\boldsymbol{\phi}_j. \tag{35}$$

This stationarity condition balances the "weighted prediction" for the $j$th coordinate against its corresponding diagonal element in $\boldsymbol{\Sigma}(\mathbf{w})^{-1}$. The equations for all $j$ couple together, much like in sparse Bayesian regression.

We next show, by contradiction, that when $\boldsymbol{\Phi}$ is well-conditioned and satisfies the RIP condition, these coupled equations admit no spurious local minima: every minimum of the loss corresponds to a global minimum. Finally, we show that in the limit $\gamma \to 0$, this global minimum converges to the sparse optimal solution.

### A.2 No Bad Local Minima for Well-Conditioned Design Matrix

**Lemma .1.** *Suppose there exist two distinct vectors $\mathbf{w}^{(1)}$ and $\mathbf{w}^{(2)}$, both satisfying the stationarity conditions over $\mathbf{x}$:*

$$\big(\boldsymbol{\phi}_j^\top\boldsymbol{\Sigma}(\mathbf{w}^{(1)})^{-1}\mathbf{x}\big)^2 \;=\; \boldsymbol{\phi}_j^\top\boldsymbol{\Sigma}(\mathbf{w}^{(1)})^{-1}\boldsymbol{\phi}_j, \quad \big(\boldsymbol{\phi}_j^\top\boldsymbol{\Sigma}(\mathbf{w}^{(2)})^{-1}\mathbf{x}\big)^2 \;=\; \boldsymbol{\phi}_j^\top\boldsymbol{\Sigma}(\mathbf{w}^{(2)})^{-1}\boldsymbol{\phi}_j, \quad \forall j. \tag{36}$$

*Then $\mathbf{w}^{(1)} = \mathbf{w}^{(2)}$ if $\boldsymbol{\Phi}$ satisfies the RIP condition with small $\delta$. In other words, there are no "bad" local minima under these stationarity conditions.*

*Proof.* The main requirement for Lemma 1 is the absence of distinct $\mathbf{w}^{(1)}$ and $\mathbf{w}^{(2)}$ that satisfy (34). With a perfect optimizer, the full rank assumption ensures the presence of a unique inverse $\boldsymbol{\Sigma}^{-1}(\mathbf{w})$ for unique $\mathbf{w}$, leading to no bad local minima. However, practical optimizers such as SGD might identify distinct $\mathbf{w}^{(1)}$ and $\mathbf{w}^{(2)}$ as minima for the loss function in (25) that satisfy $\boldsymbol{\Sigma}^{-1}(\mathbf{w}^{(1)}) \approx \boldsymbol{\Sigma}^{-1}(\mathbf{w}^{(2)})$. We use the RIP bound $\delta$ to ensure that the difference in inverse terms is large enough to be detectable by SGD.

Let $\mathbf{z} = \sum_{i \in S}\alpha_i e_i$, where $S \subseteq \{1,\ldots,n\}$ with $|S| = \kappa$, be a $\kappa$-sparse vector and let $\boldsymbol{\Phi} \in \mathbb{R}^{d\times n}$ with columns $\boldsymbol{\phi}_1,\ldots,\boldsymbol{\phi}_n$. Then, $\boldsymbol{\Phi}\mathbf{z} = \sum_{i \in S}\alpha_i\boldsymbol{\phi}_i$ and its squared norm of $\boldsymbol{\Phi}\mathbf{z}$ becomes: $\|\boldsymbol{\Phi}\mathbf{z}\|_2^2 = \big\|\sum_{i \in S}\alpha_i\boldsymbol{\phi}_i\big\|_2^2$.

A.2.1 DEVIATION FROM ISOMETRY AND RIP CONSTANT

The RIP condition for $\kappa$-sparse vectors requires:

$$(1 - \delta)\|\mathbf{z}\|_2^2 \leq \|\mathbf{\Phi z}\|_2^2 \leq (1 + \delta)\|\mathbf{z}\|_2^2 \tag{37}$$

Subtracting $\|\mathbf{z}\|_2^2$, we obtain the deviation:

$$\|\mathbf{\Phi z}\|_2^2 - \|\mathbf{z}\|_2^2 = \sum_{i \in S} \alpha_i^2(\|\phi_i\|_2^2 - 1) + \sum_{\substack{i,j \in S \\ i \neq j}} \alpha_i \alpha_j \langle \phi_i, \phi_j \rangle \tag{38}$$

Thus, the RIP constant $\delta$ is defined as the worst-case deviation over all $\kappa$-sparse unit-norm vectors $\mathbf{z}$:

$$\delta = \max_{\substack{S \subseteq \{1,\ldots,n\} \\ |S| = \kappa \\ \sum_{i \in S} \alpha_i^2 = 1}} \left| \sum_{i \in S} \alpha_i^2(\|\phi_i\|_2^2 - 1) + \sum_{\substack{i,j \in S \\ i \neq j}} \alpha_i \alpha_j \langle \phi_i, \phi_j \rangle \right| \tag{39}$$

A.2.2 RELATIONSHIP BETWEEN $\delta$ AND SGD

For a $\kappa$-sparse $\mathbf{z}$, the RIP bound $\delta$ can be expressed as a weighted sum of activated column norms of $\mathbf{\Phi}$ and cross correlations between them. However, the presence of aligned columns leads to large correlations, increasing the $\delta$ value.

Without loss of generality, pick an index $j \in \mathcal{S}_1$ but $j \notin \mathcal{S}_2$. Thus, $\mathbf{w}^{(1)}$ "turns on" column $\phi_j$ while $\mathbf{w}^{(2)}$ has $w_j^{(2)} = 0$. If $\phi_j$ is co-linear with other columns it will lead to a large $\delta$. Furthermore, it also means a small difference in $\Sigma^{-1}(\mathbf{w}^{(1)})$ and $\Sigma^{-1}(\mathbf{w}^{(2)})$ as $\mathbf{w}^{(1)}$ and $\mathbf{w}^{(2)}$ differ along the indices of the aligned columns having large correlations. It is due to the common large correlation term with $\phi_j$ both in $\Sigma^{-1}(\mathbf{w}^{(1)})$ and $\Sigma^{-1}(\mathbf{w}^{(2)})$, they will have a small difference.

Therefore small $\delta$ suggests small correlation and therefore a larger separation between $\Sigma^{-1}(\mathbf{w}^{(1)})$ and $\Sigma^{-1}(\mathbf{w}^{(2)})$. This indicates that a small $\delta$ is essential to find the true local/global optimum.

A.2.3 PROOF BY CONTRADICTION

We argue by contradiction. Assume $\mathbf{w}^{(1)} \neq \mathbf{w}^{(2)}$ while both vectors satisfy (36). Define the *active set* $\mathcal{S}_m = \{ j \mid w_j^{(m)} > 0 \}$ for $m = 1, 2$. Note that each $\mathbf{w}^{(m)}$ is strictly positive in its active coordinates, and hence corresponds to selecting a certain subset of columns from $\mathbf{\Phi}$. Our goal is to show that this situation cannot arise if $\mathbf{\Phi}$ is well-conditioned.

**Case 1: $\mathcal{S}_1 \neq \mathcal{S}_2$.** Without loss of generality, pick an index $j \in \mathcal{S}_1$ but $j \notin \mathcal{S}_2$. Thus, $\mathbf{w}^{(1)}$ "turns on" column $\phi_j$ while $\mathbf{w}^{(2)}$ has $w_j^{(2)} = 0$. Because $\mathbf{\Phi}$ satisfied the RIP condition (Candes & Tao, 2005), $\phi_j$ cannot be near-collinear with the other active columns in $\mathcal{S}_2$. Consequently, $\mathbf{\Sigma}(\mathbf{w}^{(1)})$ and $\mathbf{\Sigma}(\mathbf{w}^{(2)})$ differ in a way that prevents both from simultaneously satisfying the stationarity conditions for the same $\mathbf{x}$. This yields a contradiction, so $\mathbf{w}^{(1)}$ and $\mathbf{w}^{(2)}$ cannot both be solutions.

**Case 2: $\mathcal{S}_1 = \mathcal{S}_2$.** In this case, both $\mathbf{w}^{(1)}$ and $\mathbf{w}^{(2)}$ activate exactly the same set of columns. For each $j \in \mathcal{S}_1$, the single-observation stationarity equation admits a *unique* positive solution for $w_j$. Consequently, $\mathbf{w}^{(1)}$ and $\mathbf{w}^{(2)}$ must coincide on every active coordinate, contradicting our assumption that they are distinct.

Since neither case can consistently support two different solutions, there cannot be a bad local minimum that allows for multiple solutions of $\mathbf{w}$. All local minima must provide the same solution as the *global minimizer*, implying no spurious local minima arise when $\mathbf{\Phi}$ is well-conditioned. ∎

If $\mathbf{\Phi}$ is ill-conditioned, some columns can become nearly dependent, causing $\mathbf{\Sigma}(\mathbf{w}^{(1)})$ and $\mathbf{\Sigma}(\mathbf{w}^{(2)})$ to be nearly identical even when $\mathbf{w}^{(1)} \neq \mathbf{w}^{(2)}$. This may introduce suboptimal local minima that trap optimization algorithms. Later, we show that VAEs can be used to find the preconditioning for $\mathbf{\Phi}$ to reduce its eigenvalue spread, potentially leading to an optimal sparse solution in specific cases.

### A.3 Optimal Sparse Solution at Global Minima

**Lemma .2.** *Consider the single-data-point loss in* (24), *and let* $\gamma \to 0$. *Suppose we fix* $k - d$ *elements of* $\mathbf{w}$ *to zero, and denote by* $\mathbf{w}_d \in \mathbb{R}^d$ *the remaining nonzero elements, with* $\mathbf{\Phi}_d \in \mathbb{R}^{d \times d}$ *the corresponding columns of* $\mathbf{\Phi}$, *and* $\mathbf{z}_d = \mathbf{\Phi}_d^{-1}\mathbf{x}$. *Then any minimizer of the loss matches these nonzero coordinates of* $\mathbf{w}$ *satisfying* $\hat{\mathbf{w}}_d = \left( \mathbf{\Phi}_d^{-1}\mathbf{x} \right)^2$.

*Proof.* In the presence of non-zero $\gamma$, the optimal sparse solution requires us to solve (34) the implicit stationarity condition for which a closed form solution does not exist. We leverage Theorem 4 from (Dai & Wipf, 2019), which states that for any $\gamma > 0$, there exists a $\gamma' < \gamma$ for which the VAE loss can be reduced. Our proposed VAE architecture satisfies the conditions for Theorem 4 from (Dai & Wipf, 2019) implying that $\gamma \to 0$ leads to minimizing the VAE loss. Therefore, it is valid to evaluate the limiting value of the loss function in (24), and use it to obtain the local/ global minimum solution. When $\gamma \to 0$, the loss in (24) takes the form

$$\mathbf{x}^\top \mathbf{\Sigma}(\mathbf{w}_d)^{-1}\mathbf{x} \;=\; \mathbf{x}^\top \Big[ \big( \mathbf{\Phi}_d^\top \big)^{-1} \operatorname{diag}\!\Big( \tfrac{1}{\mathbf{w}_d} \Big) \mathbf{\Phi}_d^{-1} \Big] \mathbf{x} \;=\; \big( \mathbf{\Phi}_d^{-1}\mathbf{x} \big)^\top \operatorname{diag}\!\Big( \tfrac{1}{\mathbf{w}_d} \Big) \big( \mathbf{\Phi}_d^{-1}\mathbf{x} \big). \tag{40}$$

Defining $\mathbf{z}_d = \mathbf{\Phi}_d^{-1}\mathbf{x}$, we isolate the $d$ elements $\{z_i\}_{i=1}^d$ corresponding to the nonzero coordinates $\{w_{d,i}\}_{i=1}^d$. This gives

$$\mathbf{x}^\top \mathbf{\Sigma}(\mathbf{w}_d)^{-1}\mathbf{x} \;=\; \sum_{i=1}^d \frac{z_i^2}{w_{d,i}}. \tag{41}$$

Using the multiplicative property of determinants,

$$\big| \mathbf{\Sigma}(\mathbf{w}_d) \big| \;=\; \big| \mathbf{\Phi}_d \operatorname{diag}(\mathbf{w}_d)\, \mathbf{\Phi}_d^\top \big| \;=\; \big| \mathbf{\Phi}_d \big| \big| \operatorname{diag}(\mathbf{w}_d) \big| \big| \mathbf{\Phi}_d^\top \big| \;=\; |\mathbf{\Phi}_d|^2 \prod_{i=1}^d w_{d,i}, \tag{42}$$

$$\implies \log \big| \mathbf{\Sigma}(\mathbf{w}_d) \big| \;=\; 2\log|\mathbf{\Phi}_d| \;+\; \sum_{i=1}^d \log w_{d,i}. \tag{43}$$

The term $2\log|\mathbf{\Phi}_d|$ is a constant with respect to $w_{d,i}$ and thus does not affect minimization. Combining these, the loss is separated over the coordinates:

$$L(\mathbf{w}_d) \;=\; \sum_{i=1}^d \frac{z_i^2}{w_{d,i}} \;+\; \sum_{i=1}^d \log w_{d,i} \;+\; \text{constant}, \quad \text{where } z_i = \big( \mathbf{\Phi}_d^{-1}\mathbf{x} \big)_i. \tag{44}$$

Since the summation is separable in each $w_{d,i}$, we optimize each coordinate independently. For the $i$th term, setting the derivative to zero gives:

$$\frac{d}{dw_{d,i}} \Big( \tfrac{z_i^2}{w_{d,i}} + \log w_{d,i} \Big) \;=\; -\frac{z_i^2}{w_{d,i}^2} \;+\; \frac{1}{w_{d,i}} \;=\; 0,$$

$$\implies \hat{w}_{d,i} \;=\; z_i^2 \;=\; \big( \mathbf{\Phi}_d^{-1}\mathbf{x} \big)_i^2.$$

A second-derivative check confirms this is a global minimum, since the objective is strictly convex in each $w_{d,i}$. Consequently, the unique minimizer over $\mathbf{w}_d$ is given by $\hat{w}_{d,i} = \big( \mathbf{\Phi}_d^{-1}\mathbf{x} \big)_i^2$. ∎

Lemma .2 shows that the global optimal solution for the VAE loss in (24) recovers the sparse representation defined by the coefficients $\mathbf{z}^*$ underlying the observation $\mathbf{x}$ in (4). Moreover, according to Lemma 4.1 any global minima of a VAE with a linear encoder-decoder achieves the optimal sparse representation for $\mathbf{z}$ when $\gamma \to 0$. Therefore resulting encoder mean given by (24) also satisfies optimal reconstruction under the limiting condition of $\gamma \to 0$ leading to:

$$\mathbf{\Phi}\operatorname{diag}[\hat{\mathbf{w}}_x]\boldsymbol{\mu}_z = \mathbf{\Phi}\operatorname{diag}[\hat{\mathbf{w}}_x]^2\mathbf{\Phi}^T(\mathbf{\Phi}\operatorname{diag}[\hat{\mathbf{w}}_x]\mathbf{\Phi}^\top)^{-1}\mathbf{x} = \mathbf{x} \tag{45}$$

$$\implies \operatorname{diag}[\hat{\mathbf{w}}_x]^2\boldsymbol{\mu}_z = \operatorname{diag}[\hat{\mathbf{w}}_x]^2\mathbf{\Phi}^T(\mathbf{\Phi}\operatorname{diag}[\hat{\mathbf{w}}_x]^2\mathbf{\Phi}^\top)^{-1}\mathbf{x} = \hat{\mathbf{z}} \tag{46}$$

$$\implies \hat{\mathbf{z}} = \operatorname{diag}[\hat{\mathbf{w}}_x](\mathbf{\Phi}\operatorname{diag}[\hat{\mathbf{w}}_x])^\dagger\mathbf{x}, \tag{47}$$

where † denotes the pseudo-inverse operation. ∎

## B. Proof of Lemma 4.3

We aim to compare the condition number of $\mathbf{P}\,\mathbf{\Phi}$ with that of $\mathbf{\Phi}$. Following the SLR setting in the main-text, $\mathbf{\Phi} \in \mathbb{R}^{d\times k}$ is a full-rank matrix with $\mathrm{rank}(\mathbf{\Phi}) = d \leq \min\{d,k\}$ and preconditioner $\mathbf{P} = \left(\mathbf{\Phi}\,\mathbf{\Phi}^\top + \gamma\,\mathbf{I}\right)^{-1}$ with $\gamma > 0$ and $\mathbf{I}$ is the $d \times d$ identity matrix.

We start with computing the singular value decomposition (SVD) of $\mathbf{\Phi} = \mathbf{U}\,\mathbf{\Sigma}\,\mathbf{V}^\top$, where $\mathbf{U} \in \mathbb{R}^{d\times d}$ and $\mathbf{V} \in \mathbb{R}^{n\times d}$ have orthonormal columns, and $\mathbf{\Sigma} \in \mathbb{R}^{d\times d}$ is diagonal matrix with the $d$ eigenvalues following the order $\sigma_1 \geq \cdots \geq \sigma_d > 0$. Then $\mathbf{\Phi}\,\mathbf{\Phi}^\top = \mathbf{U}\,\mathbf{\Sigma}^2\,\mathbf{U}^\top$ which yields,

$$\mathbf{\Phi}\,\mathbf{\Phi}^\top + \gamma\,\mathbf{I} \;=\; \mathbf{U}\left(\mathbf{\Sigma}^2 + \gamma\,\mathbf{I}\right)\mathbf{U}^\top + \gamma\left(\mathbf{I} - \mathbf{U}\,\mathbf{U}^\top\right). \tag{48}$$

and hence,

$$\mathbf{P} \;=\; \left(\mathbf{\Phi}\,\mathbf{\Phi}^\top + \gamma\,\mathbf{I}\right)^{-1} \;=\; \mathbf{U}\left(\mathbf{\Sigma}^2 + \gamma\,\mathbf{I}\right)^{-1}\mathbf{U}^\top + \frac{1}{\gamma}\left(\mathbf{I} - \mathbf{U}\,\mathbf{U}^\top\right). \tag{49}$$

The preconditioned design matrix $\mathbf{P}\,\mathbf{\Phi}$ is given by,

$$\mathbf{P}\,\mathbf{\Phi} \;=\; \left[\mathbf{U}\left(\mathbf{\Sigma}^2 + \gamma\,\mathbf{I}\right)^{-1}\mathbf{U}^\top + \tfrac{1}{\gamma}\left(\mathbf{I} - \mathbf{U}\,\mathbf{U}^\top\right)\right]\left(\mathbf{U}\,\mathbf{\Sigma}\,\mathbf{V}^\top\right) \tag{50}$$

$$\implies \mathbf{P}\,\mathbf{\Phi} \;=\; \mathbf{U}\left(\mathbf{\Sigma}^2 + \gamma\,\mathbf{I}\right)^{-1}\mathbf{\Sigma}\,\mathbf{V}^\top \;=\; \mathbf{U}\,\mathbf{M}\,\mathbf{V}^\top. \tag{51}$$

where

$$\mathbf{M} \;=\; \left(\mathbf{\Sigma}^2 + \gamma\,\mathbf{I}\right)^{-1}\mathbf{\Sigma} \;=\; \mathrm{diag}\!\left(\tfrac{\sigma_1}{\sigma_1^2+\gamma},\ldots,\tfrac{\sigma_d}{\sigma_d^2+\gamma}\right). \tag{52}$$

Since $\mathbf{U}$ and $\mathbf{V}$ are orthonormal, the singular values of $\mathbf{P}\,\mathbf{\Phi}$ are the diagonal entries of $\mathbf{M}$, i.e. $\sigma_i/(\sigma_i^2 + \gamma)$ for $i = 1,\ldots,d$. Hence,

$$\sigma_{\max}(\mathbf{P}\,\mathbf{\Phi}) \;=\; \max_{1\leq i\leq d}\frac{\sigma_i}{\sigma_i^2 + \gamma}, \quad \sigma_{\min}(\mathbf{P}\,\mathbf{\Phi}) \;=\; \min_{1\leq i\leq d}\frac{\sigma_i}{\sigma_i^2 + \gamma}. \tag{53}$$

Therefore, the condition number satisfies

$$\mathrm{cond}(\mathbf{P}\,\mathbf{\Phi}) \;=\; \frac{\sigma_{\max}(\mathbf{P}\,\mathbf{\Phi})}{\sigma_{\min}(\mathbf{P}\,\mathbf{\Phi})} \;\leq\; \frac{\frac{\sigma_1}{\sigma_1^2+\gamma}}{\frac{\sigma_d}{\sigma_d^2+\gamma}} \;=\; \left(\tfrac{\sigma_1}{\sigma_d}\right)\left(\tfrac{\sigma_d^2+\gamma}{\sigma_1^2+\gamma}\right) \;=\; \mathrm{cond}(\mathbf{\Phi})\cdot\frac{\sigma_d^2(\mathbf{\Phi})+\gamma}{\sigma_1^2(\mathbf{\Phi})+\gamma}. \tag{54}$$

Since $\left(\sigma_d^2(\mathbf{\Phi}) + \gamma\right)/\left(\sigma_1^2(\mathbf{\Phi}) + \gamma\right) \leq 1$, we have

$$\boxed{\mathrm{cond}(\mathbf{P}\,\mathbf{\Phi}) \;\leq\; \mathrm{cond}(\mathbf{\Phi}).} \tag{55}$$

Equality holds only if all singular values of $\mathbf{\Phi}$ are equal; otherwise, multiplying $\mathbf{\Phi}$ on the left by $(\mathbf{\Phi}\,\mathbf{\Phi}^\top + \gamma\,\mathbf{I})^{-1}$ strictly reduces its condition number. $\blacksquare$

## C. Proof of Theorem 4.4

This proof builds on the arguments from the proof of Theorem 4.2 (see Appendix A), except for the limiting behavior of $\gamma$. Here, we do not require $\gamma \to 0$, but rather allow it to take a value which would satisfy the RIP condition. After preconditioning the design matrix $\mathbf{\Phi}$ and the observation vector $\mathbf{x}$ by $\mathbf{P} = \left(\mathbf{\Phi}\,\mathbf{\Phi}^\top + \gamma\,\mathbf{I}\right)^{-1}$, the loss in (21) becomes

$$\mathcal{L}(\theta,\phi) = \frac{1}{\gamma}\|\mathbf{P}\mathbf{x} - \mathbf{P}\mathbf{\Phi}\,\mathrm{diag}[\mathbf{w}_x]\,\mathbf{W}_z\,\mathbf{P}\mathbf{x}\|_2^2 + \frac{1}{\gamma}\mathrm{tr}\!\left[\mathrm{diag}[\mathbf{w}_x]\,\mathbf{\Phi}^\top\,\mathbf{P}^\top\mathbf{P}\mathbf{\Phi}\,\mathrm{diag}[\mathbf{w}_x]\,\mathbf{S}\,\mathbf{S}^\top\right]$$
$$+ d\log\gamma + \mathrm{tr}\!\left[\mathbf{S}\,\mathbf{S}^\top\right] - \log\left|\mathbf{S}\,\mathbf{S}^\top\right| + \|\mathbf{W}_z\,\mathbf{P}\mathbf{x}\|_2^2, \tag{56}$$

with $\theta = \{\mathbf{w}_x, \gamma\}$ and $\phi = \{\mathbf{W}_z, \mathbf{S}\}$. Replacing $\tilde{\mathbf{\Phi}} = \mathbf{P}\mathbf{\Phi}$ and $\tilde{\mathbf{x}} = \mathbf{P}\mathbf{x}$ in (56) we get:

$$\mathcal{L}(\theta, \phi) = \frac{1}{\gamma} \|\tilde{\mathbf{x}} - \tilde{\boldsymbol{\Phi}} \operatorname{diag}[\mathbf{w}_x] \mathbf{W}_z \tilde{\mathbf{x}}\|_2^2 + \frac{1}{\gamma} \operatorname{tr}\big[\operatorname{diag}[\mathbf{w}_x] \tilde{\boldsymbol{\Phi}}^\top \tilde{\boldsymbol{\Phi}} \operatorname{diag}[\mathbf{w}_x] \mathbf{S}\,\mathbf{S}^\top\big]$$
$$+ d \log \gamma + \operatorname{tr}\big[\mathbf{S}\,\mathbf{S}^\top\big] - \log\big|\mathbf{S}\,\mathbf{S}^\top\big| + \|\mathbf{W}_z \tilde{\mathbf{x}}\|_2^2. \tag{57}$$

For a fixed $\gamma$, it follows from Theorem 4.2 that (57) has no bad local minima if the RIP condition holds for $\tilde{\boldsymbol{\Phi}}$ with the chosen $\gamma$. Clearly, $\tilde{\boldsymbol{\Phi}}$ will not satisfy RIP for every $\gamma$ because $\boldsymbol{\Phi}$ itself is not guaranteed to satisfy RIP. Lemma 4.3 implies that $\operatorname{cond}(\tilde{\boldsymbol{\Phi}}) \leq \operatorname{cond}(\boldsymbol{\Phi})$, but does not assure that $\tilde{\boldsymbol{\Phi}}$ satisfies RIP for all $\gamma$.

Suppose there exists a $\gamma = \gamma^*$ such that $\tilde{\boldsymbol{\Phi}}$ satisfies RIP (i.e., its columns are linearly independent for the required support). In that case, (57) has no bad local minima, and any minimum of (57) is also a global minimum. This minimum leads to the loss-minimizing $\hat{\mathbf{w}}_x$, consistent with the stationarity conditions:

$$\mathcal{L}(\mathbf{w}) = \mathbf{x}^\top \boldsymbol{\Sigma}_x^{-1} \mathbf{x} + \log\big|\boldsymbol{\Sigma}_x\big|, \quad \text{where } \boldsymbol{\Sigma}_x \triangleq \tilde{\boldsymbol{\Phi}} \mathbf{W} \tilde{\boldsymbol{\Phi}}^\top + \gamma^* \mathbf{I}. \tag{58}$$

The uniqueness of $\gamma^*$ that induces an RIP-compliant $\tilde{\boldsymbol{\Phi}}$, and hence eliminates bad minima in (57), is not guaranteed. However, since $\tilde{\boldsymbol{\Phi}}$ meets RIP, Theorem 4.2 implies that using $\tilde{\boldsymbol{\Phi}}$ as the design matrix and $\tilde{\mathbf{x}}$ as observations, the loss in (24) can attain the optimal sparse representation as $\gamma \to 0$. Because the optimal sparse representation $\mathbf{z}^*$ is unique, the specific $\gamma^*$ that enhances the RIP condition will also converge to this optimal sparse solution if it is sufficiently small to meet the limiting conditions for $\gamma \to 0$. This holds when

$$\lim_{\gamma \to 0} \operatorname{diag}[\hat{\mathbf{w}}_x]^2 \tilde{\boldsymbol{\Phi}}^\top \big(\tilde{\boldsymbol{\Phi}} \operatorname{diag}[\hat{\mathbf{w}}_x]^2 \tilde{\boldsymbol{\Phi}}^\top + \gamma \mathbf{I}\big)^{-1} \tilde{\mathbf{x}} \;\to\; \operatorname{diag}[\hat{\mathbf{w}}_x]^2 \tilde{\boldsymbol{\Phi}}^\top \big(\tilde{\boldsymbol{\Phi}} \operatorname{diag}[\hat{\mathbf{w}}_x]^2 \tilde{\boldsymbol{\Phi}}^\top\big)^{-1} \tilde{\mathbf{x}}$$

$$\approx \; \operatorname{diag}[\hat{\mathbf{w}}_x]^2 \tilde{\boldsymbol{\Phi}}^\top \big(\tilde{\boldsymbol{\Phi}} \operatorname{diag}[\hat{\mathbf{w}}_x]^2 \tilde{\boldsymbol{\Phi}}^\top + \gamma^* \mathbf{I}\big)^{-1} \tilde{\mathbf{x}}, \tag{59}$$

where

$$\tilde{\boldsymbol{\Phi}} = \big(\boldsymbol{\Phi}\,\boldsymbol{\Phi}^\top + \gamma^* \mathbf{I}\big)^{-1} \boldsymbol{\Phi}, \quad \tilde{\mathbf{x}} = \big(\boldsymbol{\Phi}\,\boldsymbol{\Phi}^\top + \gamma^* \mathbf{I}\big)^{-1} \mathbf{x}. \tag{60}$$

Thus,

$$\hat{\mathbf{z}} = \operatorname{diag}[\hat{\mathbf{w}}_x]^2 \tilde{\boldsymbol{\Phi}}^\top \big(\tilde{\boldsymbol{\Phi}} \operatorname{diag}[\hat{\mathbf{w}}_x]^2 \tilde{\boldsymbol{\Phi}}^\top\big)^{-1} \tilde{\mathbf{x}}$$

$$= \operatorname{diag}[\hat{\mathbf{w}}_x]^2 \boldsymbol{\Phi}^\top \hat{\mathbf{P}}^\top \big(\hat{\mathbf{P}} \boldsymbol{\Phi} \operatorname{diag}[\hat{\mathbf{w}}_x]^2 \boldsymbol{\Phi}^\top \hat{\mathbf{P}}^\top\big)^{-1} \hat{\mathbf{P}}\,\mathbf{x}. \tag{61}$$

$\blacksquare$

## D. Additional Experimental Results

In this section, we provide all the additional experimental results comparing our technique with other works for different design matrices. First, summarize the methodology and key findings for each parameter variation (Section 5.4), followed by the additional results for the Riboflavin dataset (Section 5.3) and the impact of fixed vs. trainable $\gamma$ on sparse recovery (Section 5.5).

### D.1 Design Matrix Dimensions

To quantify the impact of feature dimension $n$ and number of observations $d$ on SLR support recovery, we chose the Gaussian Random Walk design matrix from Section 5.2 with sparsity level $\kappa = 20$ and $\kappa = 10$ respectively. First, we fixed the number of observations at $d = 100$ and varied the number of features $n$ from 150 to 300 in steps of 50. Then, holding $n = 200$ constant, we varied $d$ from 10 to 100. In each configuration, the nonzero coefficients were drawn i.i.d. from $\mathcal{N}(0, 1)$, and we performed 10 independent trials, measuring the fraction of trials in which the estimated support matched the ground truth exactly. As shown in Fig. 2(a), recovery performance for all methods degrades as $n$ increases (i.e., as the effective sparsity $k/n$ decreases), but our preconditioned VAE consistently achieves the highest support recovery rate across the entire range. Likewise, Fig. 2(b) demonstrates that increasing the observation count $d$ improves recovery for all algorithms, with the VAE maintaining a 5–10% advantage over LASSO, and Augmented basis pursuit at each sample size. Both observations follow the insight that as more information becomes available for solving the SLR, the support recovery rate improves, with preconditioned VAE performing better than the others.

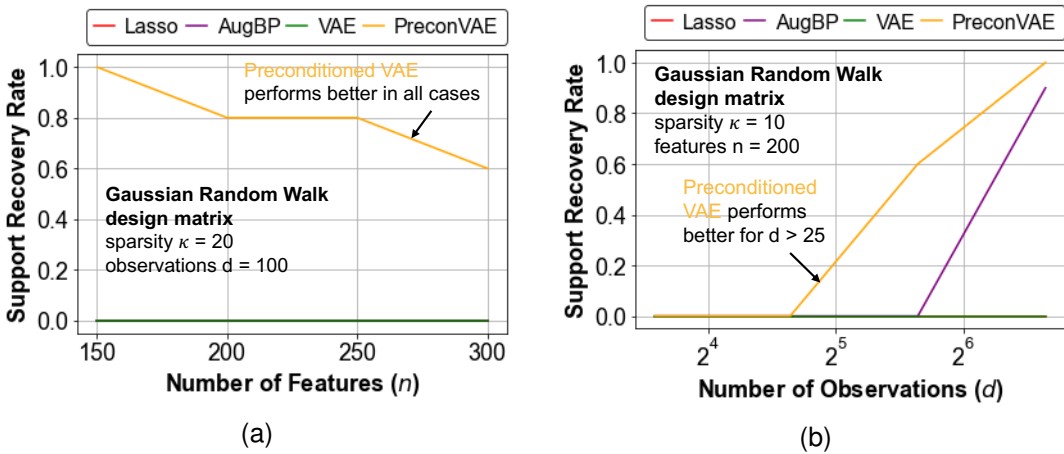

Figure 2. Sparse recovery rate vs. (a) number of features $n$, and (b) number of observations $d$ for Gaussian random walk design matrix.

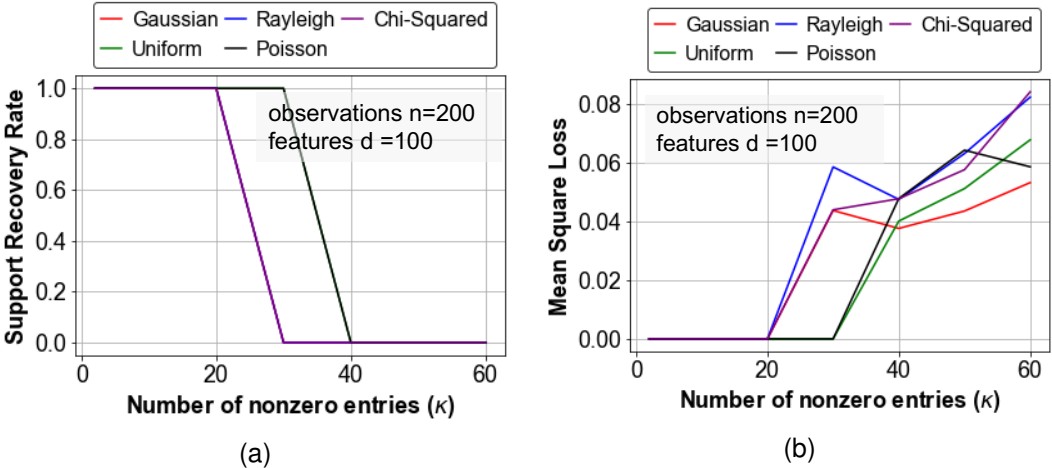

Figure 3. (a) Support recovery rate and (b) mean squared error using proposed VAE followed by standard least squares for different distributions of non-zero coefficients on standard Gaussian design matrix.

### D.2 Nonzero Coefficient Distribution

We next evaluated whether the distribution of the nonzero coefficients affects recovery. Keeping $(n, d) = (200, 100)$, we sampled the nonzero entries from five distinct distributions: the standard Gaussian, Rayleigh, Chi-squared, Uniform, and Poisson distributions. For each distribution, we ran 10 trials and recorded both the support recovery rate (Fig. 3(a)) and the mean-squared error (MSE) of the coefficient estimates (Fig. 3(b)). The MSE curves for cases overlap, and the support recovery rates differ by no more than two percentage points. This confirms that our VAE's mechanism for identifying support is essentially invariant to the actual value distribution of the nonzero coefficients.

### D.3 Noise Level

We evaluated performance under additive measurement noise. Using $(n, d) = (200, 100)$ and Gaussian distributed nonzero coefficients, we injected noise $\eta \sim \mathcal{N}(0, \sigma^2)$ and defined the signal-to-noise ratio as $\mathrm{SNR} = 10 \log_{10}\big(\mathrm{Var}(\mathbf{\Phi z})/\sigma^2\big)$. We varied $\mathrm{SNR}$ from $0\,\mathrm{dB}$ to $80\,\mathrm{dB}$, running 10 trials for each evaluation. For the standard Gaussian design matrix (Fig. 4(a)), we found that although all methods improve with increasing SNR, our VAE outperforms LASSO and SBL at low SNR ($20\,\mathrm{dB}$ to $40\,\mathrm{dB}$). Similarly, for the Gaussian random walk design matrix (Fig. 4(b)), we observed that while all methods fail at low SNR, only the preconditioned VAE succeeds at SNR above $60\,\mathrm{dB}$. These results suggest promising directions for future work, such as integrating VAE-based denoising or mixture-of-Gaussians preprocessing to extend reliable support recovery into even lower SNR regimes.

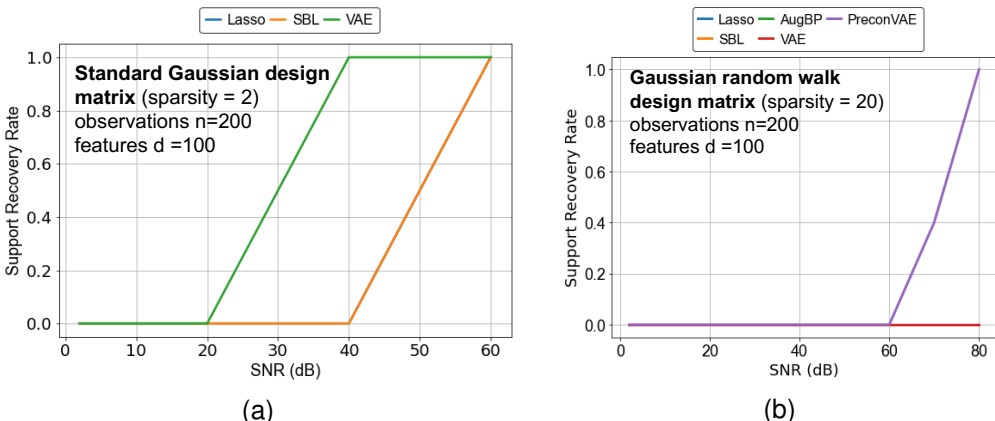

*Figure 4.* Sparse recovery rate for (a) standard Gaussian design matrix and (b) Gaussian random walk matrix vs. increasing SNR.

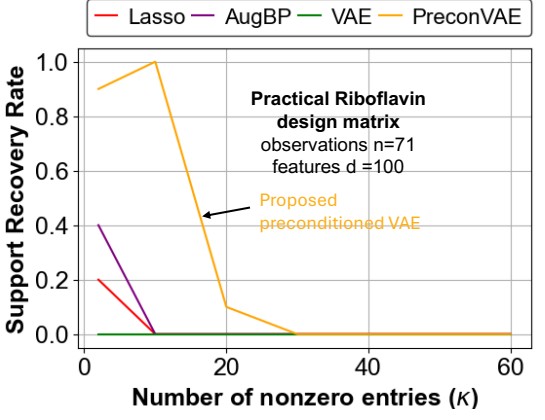

*Figure 5.* Sparse recovery rate vs. sparsity for a generalized Riboflavin matrix.

*Figure 6.* Sparse recovery rate vs. sparsity for Gaussian random walk design matrix with preconditioned VAE and (a) fixed gamma, and (b) trainable gamma.

### D.4 Additional Experiments on Riboflavin Dataset

The Riboflavin dataset used for experiments in Section 5.3 is the set of 100 genetic features exhibiting the highest empirical variances from a total of 4088 available features (Bühlmann et al., 2014). Indeed, this subset yields the worst condition number (2248). When selecting a more general subset consisting of $n = 200$ randomly chosen features, we observed an improved condition number of 1345. Notably, in this scenario with a better-conditioned matrix, our preconditioned VAE demonstrates better support recovery performance compared to others, as illustrated in Fig. 5.

### D.5 Fixed vs. Trainable $\gamma$ during Preconditioning

The trainable hyperparameter gamma significantly enhances the performance of the preconditioned VAE in addressing ill-conditioned SLR problems. According to Lemma 4.3, a positive gamma term directly improves the condition number of the design matrix. Incorporating gamma as a trainable component during optimization further facilitates superior support recovery. In Fig. 6, we examine both fixed and trainable gamma scenarios, demonstrating that the trainable gamma approach achieves higher support recovery rates. Although the VAE architecture satisfies the criterion of having no bad local minima under a fixed gamma setting (Theorem 4.2), optimal sparse recovery is contingent upon approaching the limiting value of the minima as gamma approaches zero. Imperfect optimization can hinder the achievement of this ideal scenario, as evidenced by our empirical results. Nevertheless, the proposed method outperforms LASSO and related techniques, underscoring its potential for effectively solving sparse inverse problems.

