# OpenReview forum: "Be a Goldfish: Forgetting Bad Conditioning in Sparse Linear Regression via Variational Autoencoders"
_ICML.cc/2025/Conference — ICML 2025 poster_

### Official Review · Reviewer_bPHS · 2025-03-14

**Overall Recommendation:** 3

**Summary:**

The paper studies the classic spare linear regression problem from a linear variational autoencoder (VAE) viewpoint. On the theoretical aspect, the paper provides two theorems. The first theorem guarantees that every local minimum of the VAE energy is a global minimum under a fixed variance and the global minimum achieves the optimal sparse solution when the variance converges to zero under the restricted isometry constant (RIP) condition. The second theorem is similar to the first one but with preconditioning to tackle the case when the design matrix is ill-conditioned. Numerical results show that the proposed VAE approach outperforms Lasso in some sparsity levels.

## update after rebuttal

The reviewer would like to thank the authors for conducting additional experiments on SBL and sharing the comparisons. As my main concerns regarding SBL have been answered, I have adjusted my rating from weak reject to weak accept. Please incorporate these important comparisons of SBL into the paper as it is a better baseline compared to LASSO. Lastly, the design matrix should be fat and full rank so I believe the size 500x100 mentioned by the authors might be a typo.

**Claims And Evidence:**

The VAE models considered in the paper are very restrictive. They are over simplified linear VAEs which we don’t use in practice. The models considered in the paper are basically linear regression models that can be derived without the knowledge of VAEs. The guarantee of the exact recovery still relies on the RIP conditions and the limiting scenario when the variance converges to zero. This does not explain how VAEs work in practice and how to design better VAEs.

**Essential References Not Discussed:**

Reference adequate.

**Experimental Designs Or Analyses:**

Similar to the above argument, it is hard to see the performance of the proposed method due to the lack of comparison to spare Bayesian learning approaches.

**Methods And Evaluation Criteria:**

It is expected to compare the proposed method with sparse Bayesian learning approaches such as Tipping’s and EM algorithms given the fact that they yield much better performance than Lasso. However, such comparisons are missing in the paper.

**Other Comments Or Suggestions:**

1. Notations in Line 112 and 116 are not consistent with Eq. (2).
2. The function $g$ in (6) needs to follow a set of properties to be a valid diversity measure. Perhaps the authors can briefly clarify how to design $g$.

**Other Strengths And Weaknesses:**

Strengths:
1. The paper is well-written and easy to follow.
2. The background knowledge is well presented.

Main weaknesses:
1. There are some clarity issues in the paper. In Theorem 4.2, the optimal sparse solution in shown in (15). However, it still depends on $\hat{w}_x$ and no algorithm is given to solve for $\hat{w}_x$. When the variance converges to zero, how to you solve the optimal $\hat{w}_x$? Is this still an NP-hard problem? Both Theorem 4.2 and 4.4 have the same issue.
2. The connections to VAEs and generative models are weak. These concepts seem redundant to this paper as the paper studies the optimization problems in (14) and (18). It is not clear to the reader how these findings can motivate better design of VAEs to model complex real-world datasets.
3. SBL algorithms provide much better support recovery performance than Lasso but comparison to SBL is missing. The numerical results in their current form are insufficient to determine the goodness of the proposed method.

**Questions For Authors:**

1. When the variance converges to zero, how to solve the optimal $\hat{w}_x$? Is this still an NP-hard problem?
2. How does the proposed VAE method compare to SBL approaches such as Tipping’s and EM algorithms?
3. How to train the VAE models and how to generate data?
4. What is the distribution of ground truth $z^*$ used in the simulation? Is the proposed method sensitive to distribution?
5. The SNR seems to be fixed in Section 5. Is the proposed method robust to different SNR levels?

**Relation To Broader Scientific Literature:**

The main theorems of this paper are related to the field of spare signal recovery in signal processing. On the machine learning side, the connections to VAEs and generative models are weak.

**Theoretical Claims:**

The reviewer did not verify the correctness of the proof.

---

> ### Author Rebuttal · Authors · 2025-04-01
>
> We thank the reviewer for providing feedback and suggesting insightful experiments for our work. Please find our detailed response below:
>
> ## Specific answers:
>
> 1. **Role of decoder variance γ**: We solve for optimal $\hat{\mathbf{w}}_x$ by training the proposed VAE with SGD while using γ as a training parameter. Theorems 4.2 & 4.4 show the existence of optimal sparse solutions when γ → 0, and our empirical results achieve the same for most cases except when SGD fails to attain the limiting solution. We have also addressed this point in detail as a part of our response to *Reviewer aNAM* on γ term in Theorem 4.4. We request you kindly to look at the same owing to limited #characters for the rebuttal, in case additional clarification is required.
>
> 2. **Comparison with other work**: Our experiments (Fig. https://anonymous.4open.science/r/VAE-for-SLR-3E8F/RB/R2FB.png ) reveal that while SBL methods such as Tipping’s and EM algorithm [a] surpasses LASSO in support recovery, our VAE-based approach demonstrates superior performance for Gaussian random walk design matrix. Additionally, our work provides theoretical optimality guarantees for VAEs even when dealing with ill-conditioned design matrices, a domain where SBL's performance remains largely unexplored [b]. Since SBL shares probabilistic roots with VAEs, this comparison suggests promising future research on investigating SBL's behavior under ill-conditioned design matrices and in high-sparsity regimes.
>
> 3. **Training VAEs**: To generate training data x we first select r uniformly random sparse locations within an n dimensional vector, then sample r non-zero coefficients from a standard normal distribution to obtain sparse coefficients z. Thereafter, x is generated by multiplying z with the design matrix Φ. These x values are used to train the VAE using SGD.
>
> 4. **Ground truth distribution**: Our proposed VAE focuses on locating non-zero coefficients of z, irrespective of their distribution. Once locations are determined, standard least squares retrieves exact coefficients. Our empirical results (Fig. https://anonymous.4open.science/r/VAE-for-SLR-3E8F/RB/R2FC.png ) demonstrate similar support recovery rate and mean squared error across various distributions, confirming our method's distributional invariance.
>
> 5. **Impact of noise**: To assess its impact of noise on sparse recovery, we performed extensive experiments at different SNR levels (Fig. https://anonymous.4open.science/r/VAE-for-SLR-3E8F/RB/R2FD.png ). While recovery rate increases with SNR in all cases, our proposed VAE obtains a higher recovery rate at a lower SNR compared to others. The higher recovery rate of VAE at lower SNR can be boosted further to achieve sparse recovery in presence of larger noise. This presents an interesting future research direction where preprocessing techniques such as filtering & mixture of Gaussian models [c] can be applied.
>
> ## Relation to broader scientific literature
>
> Our proposed VAE architecture focuses on theoretical exploration, beyond generative tasks like image or text synthesis. We aim to understand VAE properties relevant to challenging NP-hard problems such as SLR. Our work opens up the application space for VAEs beyond standard generative domains, such as in areas with sparsity constraints like brain anomaly localization and genome selection [d].
>
> ## Other comments or suggestions:
>
> 1. **Notations**: We are referring to the posterior probability $p_\theta(z |x)$  in Line 112 and Line 116, which is being modeled by $q_\theta(z |x)$ in Eq. (2). The likelihood term $p_\theta(x |z)$ in Eq. (2), is different from posterior $p_\theta(z |x)$ in Eq. (2). We will update the explanation in the manuscript to enhance clarity.
>
> 2. **Sparse penalty term**: In Eq. (6), g is a penalty term for the coefficients which promotes sparsity. Please note, we want a sparsity promoting term for defining a cost function for SLR instead of a diversity promoting term which increases variety in coefficients, e.g., g is L1 norm for LASSO.
>
> 3. **Choice of linear architecture**: Lemma 4.1 connects the linear VAE global minimum to optimal sparse representations. We agree with the reviewer that linear VAEs limit conventional generative capabilities such as for images and text; however, they offer valuable insights into VAE properties. Please note that standard linear regression (LR), fails for sparse, underdetermined systems ( see Fig. https://anonymous.4open.science/r/VAE-for-SLR-3E8F/RB/R2FA.png ) due to exponentially large solution space where VAEs are capable of finding the sparse solution.
>
> References:
> [a] https://arxiv.org/abs/2105.10439;
> [b] https://ieeexplore.ieee.org/document/9810291;
> [c] https://ieeexplore.ieee.org/document/9420311;
> [d] https://www.mdpi.com/1099-4300/26/9/794.

---

> > ### Comment · Reviewer_bPHS · 2025-04-05
> >
> > The reviewer would like to thank the authors for their response and for providing some results on SBL. Despite it addresses some of my concerns, more work is needed. The major concern here is that it is still unclear how the proposed VAE method compared to Tipping's or EM SBL under the standard assumption where the elements of the design matrix follow a Gaussian distribution. Also note that diversity measure minimization is a standard approach for promoting sparsity. In sparse signal recovery literature, $g$ is usually referred to as the general diversity measure and there are several properties for a separable diversity measure. The reviewer will keep the rating unchanged and encourage the authors to conduct further study on SBL and incorporate it in the paper.

---

> > > ### Author Response · Authors · 2025-04-06
> > >
> > > We thank the reviewer for the comment and our comparison details are as follows:
> > >
> > > ## **Comparison with Tipping’s algorithm under standard Gaussian distribution**
> > >
> > > In the case of the well-conditioned design matrices such as the standard Gaussian design matrix (500x100), VAEs empirically demonstrate a higher tolerance to low sparsity compared to Tipping's SBL [f] (with EM) in achieving a higher recovery rate especially with a larger number of features (n=500) (please see Fig. [https://anonymous.4open.science/r/VAE-for-SLR-3E8F/RB/R2FJ.png ]). Furthermore our VAE-based approach offers a low per epoch time complexity of $O(n^2)$ (in backpropagation through a linear VAE) compared to $O(n^3)$ for matrix inversion in EM-based SBL, where $n$ is the dimension of the sparse coefficients $z$.
> > >
> > > Note: Our preconditioned VAE achieves both higher recovery rate and tolerance to lower sparsity for ill-conditioned design matrices compared to EM-based SBL (as shown in Fig. [https://anonymous.4open.science/r/VAE-for-SLR-3E8F/RB/R2FJ.png ]). This is because the VAE approach relies on SGD for minimizing loss that does not involve matrix inversion, which on the other hand is an integral part of EM-based SBL leading to inaccurate retrieval for ill-conditioned matrices.
> > >
> > > ## **Advantages of proposed VAE over SBL-based approach**
> > > 1. **Obtaining the global minima**: The global minima for SBL in context of SLR corresponds to the optimally sparse coefficients [g] but reaching the same is not guaranteed for EM-algorithms as they can converge to a spurious local minima. Our VAE architecture smoothes out the bad local minima in the loss landscape and guarantees that reaching the global minima coincides with the optimal sparse solution in the limiting case of the decoder variance (Theorem 4.2 of the manuscript).
> > >
> > > 2. **Handling Ill-conditioned matrices**: Ill-conditioned design matrices which don’t satisfy the RIP bound, lead to increased numerical instability during matrix inversion of the EM algorithm for SBL, thereby lowering the sparse recovery rate. On the other hand our VAE algorithm preconditions the ill-conditioned design matrices to satisfy RIP, thereby achieving a higher recovery rate (Fig. H [https://anonymous.4open.science/r/VAE-for-SLR-3E8F/RB/R2FJ.png ]) compared to SBL at the same sparsity.
> > >
> > > 3. **Computation complexity**: Each iteration of the EM algorithm has the matrix inversion step of time complexity $O(n^3)$, where n is the dimension of the sparse coefficients $z$. This matrix inversion makes SBL computationally expensive, limiting the algorithm's applicability for large n. In contrast the backpropagation for our linear VAE is of the order $O(n^2)$ per epoch as it does not encounter the inverse operation.
> > >
> > > 4. **Prior information for sparse recovery**: The SBL algorithm is an empirical Bayesian method as opposed to VAE which employs variational inference. Therefore SBL, performing a type II maximum likelihood estimation, performs better with the a-priori knowledge of the distribution of non-zero coefficients in $z$ [h]. However this information is difficult to obtain in real-life settings as coefficients are unknown. VAEs fill this gap by incorporating a trainable diagonal matrix $\text{diag}(\mathbf{w}$) into the decoder architecture that captures the “sparsity” information without explicitly modeling the prior distribution. This is empirically validated (also in our previous response to Reviewer bPHS) in Fig. [https://anonymous.4open.science/r/VAE-for-SLR-3E8F/RB/R2FC.png ] showing that support recovery of VAEs is invariant to the distribution of the non-zero coefficients.
> > >
> > > Given an opportunity. we will add a new subsection in Section 4.2, to reflect the comparison with SBL techniques in detail.
> > >
> > > Lastly, we have used the ‘g’ notation for the sparsity-inducing penalty term in Eq. (6) of the manuscript based on Eq. (2) in [h] (Wipf et al., 2011).
> > >
> > > ### **References**
> > >
> > > [f] Tipping, M.E., 2001. Sparse Bayesian learning and the relevance vector machine. Journal of machine learning research, 1(Jun), pp.211-244.
> > >
> > > [g] Wipf, D.P. and Rao, B.D., 2004. Sparse Bayesian learning for basis selection. IEEE Transactions on Signal processing, 52(8), pp.2153-2164.
> > >
> > > [h] Wipf, D.P., Rao, B.D. and Nagarajan, S., 2011. Latent variable Bayesian models for promoting sparsity. IEEE Transactions on Information Theory, 57(9), pp.6236-6255.

---

### Official Review · Reviewer_7yVH · 2025-03-17

**Overall Recommendation:** 3

**Summary:**

This paper studies  the application of Variational Autoencoders for the sparse linear regression problem, particularly in cases involving ill-conditioned design matrices and low sparsity, situations where traditional methods like LASSO often fail.

**Claims And Evidence:**

The claims in the paper are supported by both theoretical analysis and empirical evidence. Specifically:
- Theoretical claims: Proved including the elimination of spurious local minima (Theorems 4.2 and 4.4) under specific architectural constraints.
- Empirical claims: Experiments conducted with three different types of design matrices (standard Gaussian, Gaussian random walk, and real-world Riboflavin dataset) demonstrate the effectiveness and improved robustness of the proposed method.

**Essential References Not Discussed:**

nothing as far as I know

**Experimental Designs Or Analyses:**

The experimental design and analyses were well-executed and sound. Specific checks included:

**Methods And Evaluation Criteria:**

The proposed VAE methods and evaluation criteria (sparse support recovery rates across various sparsity levels and design matrix conditions) are sensible and appropriately chosen for the problem context.

**Other Comments Or Suggestions:**

Minor stylistic corrections could further enhance clarity (e.g., consistency in variable definitions, notation simplicity).

**Other Strengths And Weaknesses:**

The work proposed to use VAE models to address solve the SLR problem. The work is self-contained interesting and sound in its own problem setup.

**Questions For Authors:**

NA

**Relation To Broader Scientific Literature:**

It extends recent theoretical results by Wipf (2023) by applying VAEs beyond Simultaneous Sparse Regression (SSR) to the more challenging scenario of Sparse Linear Regression (SLR).

**Theoretical Claims:**

No significant issues were found with the proofs provided.

---

> ### Author Rebuttal · Authors · 2025-04-01
>
> We are grateful to the reviewer for taking the time to review this work and provide valuable comments. We will be sure to update our manuscript to correct for stylistic corrections. Some of the plausible modifications include:
>
> 1. Using the same variable “n” for the coefficient dimension of SLR design matrix and latent dimension in VAE.
> 2. Hat (local/ global minimum) and star (ground truth) notations can be reserved for the theorem/ lemma results, and all intermediate variables can be represented without hat and star. We will clarify this in the text about specific variable usage and update with the regular notations in the manuscript.

---

### Official Review · Reviewer_aNAM · 2025-03-26

**Overall Recommendation:** 3

**Summary:**

The paper addresses the benefits of the variational autoencoder (VAE) objective to solve sparse linear regression (SLR). It shows that a particular VAE setup solves SLR under a restricted isoperimetry property (RIP) condition (Theorem 4.2), and a modification of this setup can induce preconditioning of the design matrix (Theorem). The benefit of this approach with respect to LASSO and augmented basis pursuit are demonstrated in an empirical validation study, specifically in the ill-conditioned and low-sparsity settings.

**Claims And Evidence:**

Claims are supported by theoretical and experimental results.
Theory supports the claim that the chosen VAE setups are well-behaved for the SLR problem, and provide conditions for such behavior (notably on decoder noise parameter gamma). See below for a detailed discussion.
However the theory only covers some properties of the theoretical optimum of the VAE objective and thus does not cover imperfect optimization.
Therefore the empirical study of section 5 is welcome, and shows some benefits of VAEs for support recovery. However, other metrics, such as bias of the estimated solution, is not investigated empirically, nor even discussed.

**Essential References Not Discussed:**

I am not sure it should be labeled as essential, but bias issues of SLR are surprisingly not discussed nor quantified, while there is an extensive literature on it notably for LASSO. E.g.

 A. Javanmard and A. Montanari. Debiasing the Lasso: optimal sample size
for Gaussian designs. Annals of Statistics, 46:2593–2622, 2018.

**Experimental Designs Or Analyses:**

The experimental design is satisfactory, baring uncertainties on the choice of hyperparameters mentioned above.

**Methods And Evaluation Criteria:**

The methods and evaluation criteria make sense however they are limited, when it comes to empirical validation.
Notably, the influence of the number of features and number of observations is not studied. Importantly, the choice/learning of decoder hyperparameter gamma is also not studied while it is crucial to the theory.

Ideally, one would also like to see the performance of the method on a real-world dataset.

**Other Comments Or Suggestions:**

The setting could be better introduced to a general audience by being explicit about what the variables and parameter represent in a standard regression setting, at the beginning of section 3

**Other Strengths And Weaknesses:**

The writing can be improved in several places (e.g. sentences with missing words).

**Questions For Authors:**

Please address my comments on:
- potential gaps/issues in the proof of theorem 4.2
- the claim that gamma can be automatically learned in theorem 4.4
- the missing analyses and metrics in the experimental section.

I hope some of the raised issues are misunderstandings, therefore I am leaning towards accept for now, but could change my score.

# Post rebuttal update
I am happy with the answers provided by the authors.

**Relation To Broader Scientific Literature:**

As far as I know, the paper mainly relates to (Wipf, 2023) addressing the problem of solving Simultaneous Sparse Regression (SSR) with VAEs. In contrast to it, the present paper addresses SLR (a more classic setting), but specifically investigates the benefits of VAEs in the challenging case of ill-conditioning and weak sparsity. This case has been address with a different approach in (Kelner et al., 2024), which is used as baseline.

**Theoretical Claims:**

I checked at a high level:
- theorem 4.2: the steps of the proof look reasonable, except for:
	-  the use of the RIP condition (section A.2 in appendix): the exact role of delta (e.g. how small it must be) is not explicit, neither in the statement of the theorem, nor in the proof. A reference to (Candes and Tao, 2005) is provided, but it is overall unsatisfactory as a proof argument, because no explicit result is given.
	- the study of the convergence of the solution as gamma goes to zero: the authors seem to take the limit of the loss and study the resulting solution, instead of proving that the solution for a fixed gamma converges itself as gamma goes to zero.
- lemma 4.3.: the steps of the proof look reasonable
- theorem  4.4: I could not check (yet). However, I could not follow in the first place the explanations associated to the main text statement: below equation (19) (line 326) it is said "treating gamma as trainable parameters effectively penalizes the condition number of Phi in the VAI objective (18)", I do not see why (I don't see the condition number appearing in the objective).

---

> ### Author Rebuttal · Authors · 2025-04-01
>
> We are grateful to the reviewer for their insightful comments on our submission. Please find our detailed response below:
>
> ## Addressing concerns about Theorem 4.2:
>
> 1. **Use of RIP condition**: Thanks for this excellent observation. The main requirement for A.2 Lemma 1 is the absence of distinct $\mathbf{w}^{(1)}$ and $\mathbf{w}^{(2)}$ that satisfy Eq. (37). With a perfect optimizer, a full rank Φ ensures the presence of a unique inverse $\Sigma^{-1}(\mathbf{w})$ for each $\mathbf{w}$, leading to no bad local minima. However, practical SGD might identify distinct $\mathbf{w}$’s with close inverse values satisfying Eq. (37). We use the RIP bound 𝛅 to ensure that the difference in inverse terms large enough to be detectable by SGD. For a 𝛋-sparse z, the RIP bound 𝛅 can be expressed as a weighted sum of activated column norms of Φ and cross correlations between them (for detailed explanation, please see: https://anonymous.4open.science/r/VAE-for-SLR-3E8F/RB/R3Exp.pdf ). However, the presence of aligned columns leads to large correlations increasing 𝛅 value. A large 𝛅 suggests small difference in the inverse terms for $\mathbf{w}$’s that differ along the indices of the aligned columns having large correlations. Therefore small 𝛅 suggests higher success rate for the optimization algorithm in finding the true local/global optimum.
> 2. **Study of convergence as gamma goes to zero**: In the presence of γ (decoder variance), the optimal sparse solution requires us to solve the implicit stationarity condition in Eq. (37) for which a closed form solution does not exist. We leverage Theorem 4 from (Dai and Wipf, 2019 [e]) which states that for any γ>0, there exists a γ′<γ  for which the VAE loss can be reduced. Our proposed VAE architecture satisfies the conditions for Theorem 4 from [e] implying that γ → 0 leads to minimizing the VAE loss. Therefore, it's valid to evaluate the limiting value of loss function in Eq. (26), and use it to obtain the local/ global minimum solution.
>
> ## γ term in Theorem 4.4
>
> The γ term is a training parameter in Theorem 4.4, and it impacts the condition number of PΦ. As shown in Lemma 4.3 any positive value of γ improves the condition number of PΦ compared to Φ. Therefore, adding γ to the overall loss function improves the effective condition number of Φ, pushing it to satisfy the RIP property.
>
> **Choice/Learning of Decoder hyper-parameter γ**: The trainable γ assists in achieving a higher support recovery rate of the preconditioned VAE for ill-conditioned SLR (Fig. [https://anonymous.4open.science/r/VAE-for-SLR-3E8F/RB/R3FH.png ]) compared to fixed γ. Although proposed VAE architecture can achieve no bad local minima condition for a fixed γ, optimal sparse recovery with γ → 0 is contingent on the success of the optimization algorithm. Imperfect optimization can hinder achieving this ideal scenario, as evidenced by our empirical results. We will update our manuscript to clarify this point.
>
> ## Experiments:
> We chose the ill-conditioned Gaussian random walk design matrix scenario from Sec. 5 to perform the following experiments:
>
> 1. Dependence on number of features: Fig. [https://anonymous.4open.science/r/VAE-for-SLR-3E8F/RB/R3FF.png ] shows that the pre-conditioned VAE consistently outperforms other methods in achieving a higher support recovery rate as the number of features increases.
> 2. Dependence on number of observations: Fig. [https://anonymous.4open.science/r/VAE-for-SLR-3E8F/RB/R3FG.png ] shows that the support recovery rate for preconditioned VAE increases with the number of observations, as with increase in observations d, more information is available for solving the SLR.
>
> Note: We performed experiments with real-world Riboflavin dataset in Sec. 5.3 using the 100 highest variances genetic features. Additional experimental results with random Riboflavin features (Fig. [https://anonymous.4open.science/r/VAE-for-SLR-3E8F/RB/R3FI.png ]) shows similar support recovery rate as shown in Sec. 5.3. These results will be added to the manuscript.
>
> ## Bias of the estimated solution
>
> We emphasize that our VAE-based sparse recovery aims to identify the positions of non-zero coefficients rather than directly estimating their values. Once the correct support (non-zero coefficient positions) is identified, the SLR problem simplifies significantly and can be solved using ordinary least squares (OLS). Under a full-rank design matrix and Gauss-Markov assumptions, OLS is an unbiased estimator leading to optimal sparse recovery with no bias. However, at high sparsity VAE achieves a suboptimal support identification, introducing a bias (Fig. [https://anonymous.4open.science/r/VAE-for-SLR-3E8F/RB/R3FE.png ]) but VAE still achieves a lower bias compared to LASSO at all sparsities.
>
> Response to other comments: We will start Section 3 with standard regression setting in the camera-ready manuscript.
>
> References: [e] https://arxiv.org/abs/1903.05789

---

### Decision · Program_Chairs · 2025-05-01

**Decision:**

Accept (poster)

**Comment:**

The reviewers overall appreciated the value of this theoretical contribution supported by experiments. The authors satisfactory answered the reviewers requests and I therefore recommend acceptance with the request to incorporate the results provided during the discussion period.